# Native American genetic ancestry and pigmentation allele contributions to skin color in a Caribbean population

Khai C Ang[1,2]*, Victor A Canfield[1,2], Tiffany C Foster[1,2], Thaddeus D Harbaugh[1,2], Kathryn A Early[1,2], Rachel L Harter[1], Katherine P Reid[1,2], Shou Ling Leong[3], Yuka Kawasawa[4,5,6], Dajiang Liu[4,7], John W Hawley[8], Keith C Cheng[1,2,4,5]*

[1]Department of Pathology, Penn State College of Medicine, Hershey, United States; [2]Jake Gittlen Laboratories for Cancer Research, Penn State College of Medicine, Hershey, United States; [3]Department of Family & Community Medicine, Penn State College of Medicine, Hershey, United States; [4]Department of Biochemistry and Molecular Biology, Penn State College of Medicine, Hershey, United States; [5]Department of Pharmacology, Penn State College of Medicine, Hershey, United States; [6]Institute of Personalized Medicine, Penn State College of Medicine, Hershey, United States; [7]Department of Public Health Sciences, Penn State College of Medicine, Hershey, United States; [8]Salybia Mission Project, Saint David Parish, Dominica

*For correspondence:
kca2@psu.edu (KCA);
kcheng76@gmail.com (KCC)

Competing interest: The authors declare that no competing interests exist.

**Abstract** Our interest in the genetic basis of skin color variation between populations led us to seek a Native American population with genetically African admixture but low frequency of European light skin alleles. Analysis of 458 genomes from individuals residing in the Kalinago Territory of the Commonwealth of Dominica showed approximately 55% Native American, 32% African, and 12% European genetic ancestry, the highest Native American genetic ancestry among Caribbean populations to date. Skin pigmentation ranged from 20 to 80 melanin units, averaging 46. Three albino individuals were determined to be homozygous for a causative multi-nucleotide polymorphism $OCA2^{NW273KV}$ contained within a haplotype of African origin; its allele frequency was 0.03 and single allele effect size was –8 melanin units. Derived allele frequencies of $SLC24A5^{A111T}$ and $SLC45A2^{L374F}$ were 0.14 and 0.06, with single allele effect sizes of –6 and –4, respectively. Native American genetic ancestry by itself reduced pigmentation by more than 20 melanin units (range 24–29). The responsible hypopigmenting genetic variants remain to be identified, since none of the published polymorphisms predicted in prior literature to affect skin color in Native Americans caused detectable hypopigmentation in the Kalinago.

## Editor's evaluation

This pigmentation study focuses on a community from Kalinago Territory from the Caribbean islands that on average possess high percentages of Indigenous American ancestry, and broadens the effort of quantifying the genetic effects on skin pigmentation in humans. This paper describes an analysis of the genetic structure of the Kalinago population in the Commonwealth of Dominica, and the relationship between ancestry and skin pigmentation in that population. They provide valuable new insights into the skin-lightening effect of Native American alleles, which likely have been obscured by the effect of European alleles in previous studies of admixed Native American populations. Additionally, this paper provides an interesting analysis of previously reported albinism alleles, which paints a more complex picture of the genetic architecture of pigmentation.

**eLife digest** The variation in skin colour of modern humans is a product of thousands of years of natural selection. All human ancestry can be traced back to African populations, which were dark-skinned to protect them from the intense UV rays of the sun.

Over time, humans spread to other parts of the world, and people in the northern latitudes with lower UV developed lighter skin through natural selection. This was likely driven by a need for vitamin D, which requires UV rays for production.

Separate genetic mechanisms were involved in the evolution of lighter skin in each of the two main branches of human migration: the European branch (which includes peoples on the Indian subcontinent and Europe) and the East Asian branch (which includes East Asia and the Americas).

A variant of the gene *SLC24A5* is the primary contributor to lighter skin colour in the European branch, but a corresponding variant driving light skin colour evolution in the East Asian branch remains to be identified.

One obstacle to finding such variants is the high prevalence of European ancestry in most people groups, which makes it difficult to separate the influence of European genes from those of other populations. To overcome this issue, Ang et al. studied a population that had a high proportion of Native American and African ancestors, but a relatively small proportion of European ancestors, the Kalinago people. The Kalinago live on the island of Dominica, one of the last Caribbean islands to be colonised by Europeans.

Ang et al. were able to collect hundreds of skin pigmentation measurements and DNA samples of the Kalinago, to trace the effect of Native American ancestry on skin colour. Genetic analysis confirmed their oral history records of primarily Native American (55%) – one of the highest of any Caribbean population studied to date – compared with African (32%) and European (12%) ancestries.

Native American ancestry had the highest effect on pigmentation and reduced it by more than 20 melanin units, while the European mutations in the genes *SLC24A5* and *SLC45A2* and an African gene variant for albinism only contributed 5, 4 and 8 melanin units, respectively. However, none of the so far published gene candidates responsible for skin lightening in Native Americans caused a detectable effect. Therefore, the gene responsible for lighter skin in Native Americans/East Asians has yet to be identified.

The work of Ang et al. represents an important step in deciphering the genetic basis of lighter skin colour in Native Americans or East Asians. A better understanding of the genetics of skin pigmentation may help to identify why, for example, East Asians are less susceptible to melanoma than Europeans, despite both having a lighter skin colour. It may also further acceptance of how variations in human skin tones are the result of human migration, random genetic variation, and natural selection for pigmentation in different solar environments.

## Introduction

Human skin pigmentation is a polygenic trait that is influenced by health and environment (*Barsh, 2003*). Lighter skin is most common in populations adapted to northern latitudes characterized by lower UV incidence than equatorial latitudes (*Jablonski and Chaplin, 2000*). Selection for lighter skin, biochemically driven by a solar UV-dependent photoactivation step in the formation of vitamin D (*Engelsen, 2010*; *Hanel and Carlberg, 2020*; *Holick, 1981*; *Loomis, 1967*) is regarded as the most likely basis for a convergent evolution of lighter skin color in European and East Asian/Native American populations (*Lamason et al., 2005*; *Norton et al., 2007*). The hypopigmentation polymorphisms of greatest significance in Europeans have two key characteristics: large effect size and near fixation. For example, the *A111T* allele in *SLC24A5* (*Lamason et al., 2005*) explains at least 25% of the difference in skin color between people of African vs. European genetic ancestry, and is nearly fixed in European populations. No equivalent polymorphism in Native Americans or East Asians has been found to date.

Native Americans share common genetic ancestry with East Asians (*Derenko et al., 2010*; *Tamm et al., 2007*), diverging before ~15 kya (*Gravel et al., 2013*; *Moreno-Mayar et al., 2018*; *Reich et al., 2012*), but the extent to which these populations share pigmentation variants remains to be determined. The derived alleles of rs2333857 and rs6917661 near *OPRM1*, and rs12668421 and

rs11238349 in *EGFR* are near fixation in some Native American populations, but all also have a high frequency in Europeans (*Quillen et al., 2012*), and none reach genome-wide significance in *Adhikari et al., 2019*. However, the latter found a significant association for the *Y182H* variant of *MSFD12* with skin color, but its frequencies were only 0.27 and 0.17 in Native Americans and East Asians, respectively, suggesting that it can explain only a small portion of the difference between Native American and/or East Asians and African skin color. Thus, the genetic basis for lighter skin pigmentation specific to Native American and East Asian populations, whose African alleles would be expected to be ancestral, remains to be found.

The shared genetic ancestry of East Asians and Native Americans suggests the likelihood that some light skin color alleles are shared between these populations. This is particularly the case for any variants that achieved fixation in their common ancestors. For Native American populations migrating from Beringia to the Tropics, selection for darker skin color also appears likely (*Jablonski and Chaplin, 2000*; *Quillen et al., 2019*). This would have increased the frequency of novel dark skin variants, if any, and would have decreased the frequency of light skin variants that had not achieved fixation. Hypopigmenting alleles are associated with the European admixture characteristic of many current Native American populations (*Brown et al., 2017*; *Gravel et al., 2013*; *Keith et al., 2021*; *Klimentidis et al., 2009*; *Reich et al., 2012*). Since the European hypopigmenting alleles may mask the effects of East Asian and Native American alleles, we searched for an admixed Native American population with high African, but low European admixture.

Prior to European contact, the Caribbean islands were inhabited by populations who migrated from the northern coast of South America (*Benn-Torres et al., 2008*; *Harvey et al., 1969*; *Honychurch, 2012*; *Island Caribs, 2016*; *Benn Torres et al., 2015*). During the Colonial period, large numbers of Africans were introduced into the Caribbean as slave labor (*Honychurch, 2012*; *Benn Torres et al., 2013*). As a consequence of African and European admixture and high mortality among the indigenous populations, Native American genetic ancestry now contributes only a minor portion (<15%) of the genetic ancestry of most Caribbean islanders (*Auton et al., 2015*; *Benn Torres et al., 2015*). The islands of Dominica and St. Vincent were the last colonized by Europeans in the late 1700s (*Honychurch, 2012*; *Honychurch, 1998*; *Rogoziński, 2000*). In 1903, the British granted 15 km² (3700 acres) on the eastern coast of Dominica as a reservation for the Kalinago, who were then called 'Carib.' When Dominica gained Independence in 1978, legal rights and a degree of protection from assimilation were gained by the inhabitants of the Carib Reserve (*Honychurch, 2012*) (redesignated *Kalinago Territory* in 2015). Oral history and beliefs among the Kalinago, numbering about 3000 living within the *Territory, 2021*; *Figure 1—figure supplement 2* are consistent with the primarily Native American and African genetic ancestry, assessed and confirmed genetically here.

Early in our genetic and phenotypic survey of the Kalinago, we noted an albino individual, and upon further investigation, we learned of two others residing in the Territory. We set out to identify the mutant albinism allele to avoid single albino allele effects that would potentially mask Native American hypopigmentation alleles. Oculocutaneous albinism (OCA) is a recessive trait characterized by visual system abnormalities and hypopigmentation of skin, hair, and eyes (*Gargiulo et al., 2011*; *Grønskov et al., 2007*; *Grønskov et al., 2014*; *Hong et al., 2006*; *Vogel et al., 2008*) that is caused by mutations in any of a number of autosomal pigmentation genes (*Carrasco et al., 2009*; *Edwards et al., 2010*; *Gao et al., 2017*; *Grønskov et al., 2013*; *Kausar et al., 2013*; *King et al., 2003*; *Spritz et al., 1995*; *Stevens et al., 1997*; *Stevens et al., 1995*; *Vogel et al., 2008*; *Woolf, 2005*; *Yi et al., 2003*). The incidence of albinism is ~1:20,000 in populations of European descent, but much higher in some populations, including many in sub-Saharan Africa (1:5000)(*Greaves, 2014*). Here, we report on the genetic ancestry of a population sample representing 15% of the Kalinago population of Dominica, the identification of the new albinism allele in that population, and measurement of the hypopigmenting effects of the responsible albinism allele, the European *SLC24A5*$^{A111T}$ and *SLC45A2*$^{L374}$ alleles. Native American genetic ancestry alone caused a measurable effect on pigmentation. In contrast, alleles identified in past studies of Native American skin color caused no significant effect on skin color.

## Results and discussion

Our search for a population admixed for Native American/African ancestries with minimal European admixture led us to the 'Carib' population in the Commonwealth of Dominica. Observations from

an initial trip to Dominica suggested wide variation in Kalinago skin color. Pursuit of the genetic studies described here required learning about oral and written histories, detailed discussion with community leadership, IRB approval from Ross University (until Hurricane Maria in 2017, the largest medical school in Dominica) and the Department of Health of the Commonwealth of Dominica, and relationship-building with three administrations of the Kalinago Council over 15 years.

## Population sample

Our DNA and skin color sampling program encompassed 458 individuals, representing 15% of the population of the territory and all three known albino individuals. Ages ranged from 6 to 93 (*Appendix 1—table 1* and *Figure 1—figure supplement 3*). We were able to obtain genealogical information for about half of the parents (243 mothers and 194 fathers). Community-defined ancestry (described as 'Black,' 'Kalinago,' or 'Mixed') for both parents was obtained for 426 individuals (92% of sample), including 108 parents from whom DNA samples were obtained (72 Kalinago, 36 Mixed, and 0 Black). They described themselves as Black, Kalinago, or Mixed from their perceived understanding of their parents or grandparents skin color.

## Kalinago genetic ancestry

The earliest western mention of the Kalinago (originally as 'Caribs') was in Christopher Columbus's journal dated November 26, 1492 (*Honychurch, 2012*). Little is known about the detailed cultural and genetic similarities and differences between them and other Caribbean pre-contact groups such as the Taino. African admixture in the present Kalinago population derived from the African slave trade; despite inquiry across community, governmental, and historical sources, we were unable to find documentation of specific regions of origin in Africa or well-defined contributions from other groups. The population's linguistics are uninformative, as they speak, in addition to English, the same French-based Antillean Creole spoken on the neighboring islands of Guadeloupe and Martinique.

To study Kalinago population structure, we analyzed an aggregate of our Kalinago SNP genotype data and HGDP data (*Li et al., 2008*) using ADMIXTURE (*Figure 1* and *Figure 1—figure supplement 1*) as described in Materials and methods. At K=3, the ADMIXTURE result confirmed the three major clusters, corresponding roughly to Africans (black cluster), European/Middle Easterners/Central and South Asians (yellow cluster), and East Asians/Native Americans (green cluster). At K=4 and higher, the red component that predominates Native Americans separates the Kalinago from the East Asians (green cluster). Consistent with prior work (*Li et al., 2008*), a purple cluster (Oceanians) appears at K=5 and a brown cluster (Central and South Asians) appears at K=6; both are minor sources of genetic ancestry in our Kalinago sample (average <1%) (*Appendix 1—table 2*).

At K=4 to K=6, the Kalinago show on average 55% Native American, 32% African, and 11–12% European genetic ancestry. Estimates from a two-stage admixture analysis are similar, as are results from local genetic ancestry analysis (see Materials and methods) (*Appendix 1—table 3*), leading to estimates of 54–56% Native American, 31–33% African, and 11–13% European genetic ancestry. The individual with the least admixture has approximately 94% Native American and 6% African genetic ancestry. The results of the principal component (PC) analysis (PCA) (*Figure 2—figure supplement 1*) were consistent with ADMIXTURE analysis. The first two PCs suggest that most Kalinago individuals show admixture between Native American and African genetic ancestry, with a smaller but highly variable European contribution apparent in the displacement in PC2 (*Figure 2—figure supplement 1*). A smaller number of Kalinago individuals with substantial East Asian genetic ancestry exhibit displacement in PC3 (*Figure 2—figure supplement 1*).

Our analysis of Kalinago genetic ancestry revealed considerably more Native American and less European genetic ancestry than the Caribbean samples of *Benn Torres et al., 2013*, and the admixed populations from the 1000 Genomes Project (1KGP) (*Auton et al., 2015*; *Figure 2*). Some Western Hemisphere Native Americans reported in *Reich et al., 2012*, have varying proportions of European but very little African admixture (*Figure 2B*). Overall, the Kalinago have more Native American and less European genetic ancestry than any other Caribbean population.

The 55% Native American genetic ancestry calculated from autosomal genotype in the Kalinago is greater than the reported 13% in Puerto Rico (*Gravel et al., 2013*), 10–15% for Tainos across the Caribbean (*Schroeder et al., 2018*), and 8% for Cubans (*Marcheco-Teruel et al., 2014*). This is also considerably higher than the reported 6% Native American genetic ancestry found in Bwa Mawego, a

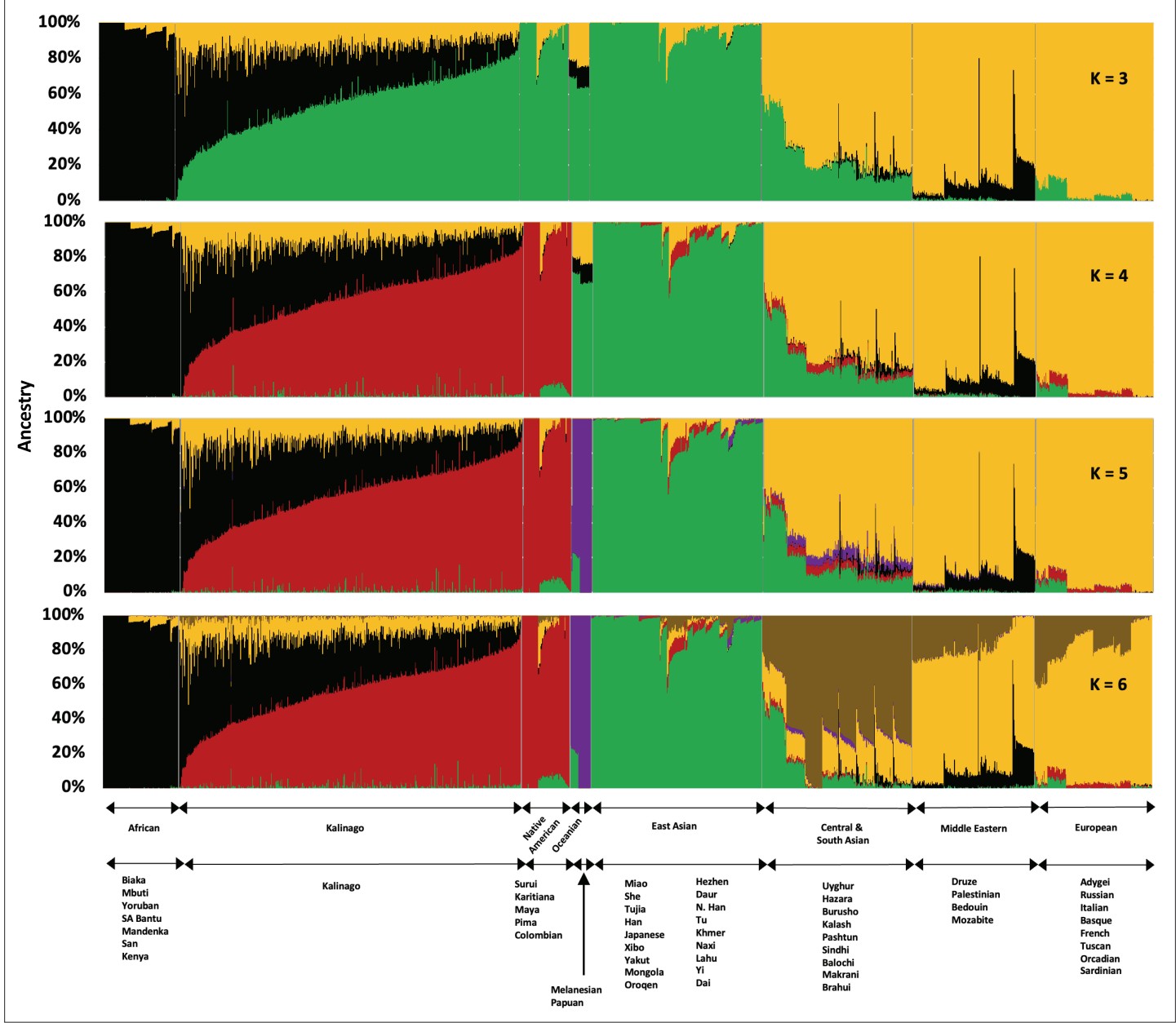

**Figure 1.** Admixture analysis of Kalinago compared with Human Genome Diversity Project populations. Results are depicted using stacked bar plots, with one column per individual. At K=3, the Kalinago, Native Americans, Oceanians, and East Asians fall into the same green cluster. At K=4, the Native Americans (red cluster) are separated from the East Asians (green cluster). *Figure 1—figure supplement 1* shows the expanded admixture plot for K=6 with each populations labeled. *Figure 1—figure supplement 2* shows the location of Kalinago Territory where fieldwork was performed.

The online version of this article includes the following source data and figure supplement(s) for figure 1:

**Source data 1.** The source data contains results from Admixture analysis.

**Figure supplement 1.** Admixture plot of Kalinago compared to Human Genome Diversity Project data from K=3 to K=6.

**Figure supplement 2.** Map showing the location of Kalinago Territory in the Commonwealth of Dominica.

**Figure supplement 3.** Age distribution of sampled Kalinago individuals.

horticultural population that resides south of the Kalinago Territory (*Keith et al., 2021*). However, this result is lower than the 67% Native American genetic ancestry reported by *Crawford et al., 2021*, for an independently collected Kalinago samples based on the mtDNA haplotype analysis. This difference suggests a paternal bias in combined European and/or African admixture. Since our Illumina SNP-chip

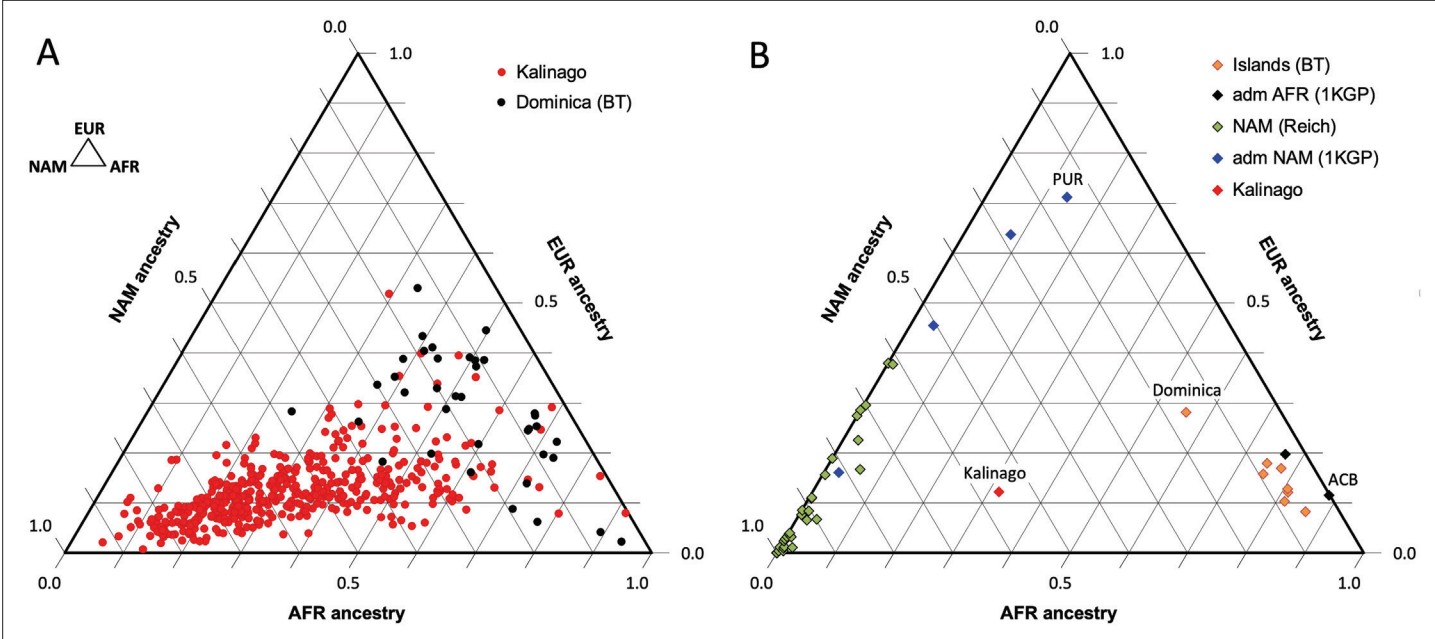

**Figure 2.** Comparison of Kalinago genetic ancestry with that of other populations in the Western Hemisphere. Ternary plots of genetic ancestry from our work and the literature show estimated proportions of African (AFR), European (EUR), and Native American (NAM) genetic ancestry. (**A**) Comparison of individuals (n=452, omitting 6 individuals with EAS >0.1) genotyped in this study to individuals (n=38) from southern Dominica sampled by *Benn Torres et al., 2013*. (**B**) Comparison of the Kalinago average genetic ancestry with other Native American populations. Kalinago, this study (n=458); Islands (BT) indicates Caribbean islanders reported in *Benn Torres et al., 2013*, with Dominica labeled; admixed (adm) AFR (1000 Genomes Project [1KGP]) and admixed NAM (1KGP) represent admixed populations from *Auton et al., 2015*, with Caribbean samples PUR (Puerto Rico) and ACB (Barbados) labeled; and AMR (Reich) indicates mainland Native American samples reported in *Reich et al., 2012*. Inset (top left) shows ancestries at vertices.

The online version of this article includes the following source data and figure supplement(s) for figure 2:

**Source data 1.** Source data contains result from PCA analysis for Kalinago versus other Native American populations in the Western Hemisphere.

**Figure supplement 1.** Principal components (PCs) analysis (PCA) of Kalinago and comparison populations.

**Figure supplement 2.** Genetic ancestry distribution as function of community-defined ancestry.

genotyping does not yield reliable identification of mtDNA haplotypes, we are currently unable to compare maternal to autosomal genetic ancestry proportions for our sample. Samples genotyped using 105 genetic ancestry informative markers from Jamaica and the Lesser Antilles (*Benn Torres et al., 2015*) yielded an average of 7.7% Native American genetic ancestry (range 5.6%–16.2%), with the highest value from a population in Dominica sampled outside the Kalinago reservation. Relevant to the potential mapping of Native American light skin color alleles, the Kalinago population has among the lowest European genetic ancestry (12%) compared to other reported Caribbean Native Americans in St. Kitts (8.2%), Barbados (11.5%), and Puerto Rico (71%) (*Benn Torres et al., 2013*). Contributing to the high percentage of Native American genetic ancestry in the Kalinago is their segregation within the 3700 acre Kalinago Territory in Dominica granted by the British in 1903, and the Kalinago tradition that women marrying non-Kalinago are required to leave the Territory; non-Kalinago spouses of Kalinago men are allowed to move to the Territory (KCA, KCC, Personal Communication with Kalinago Council, 2014). These factors help to explain why samples collected outside the Kalinago Territory (*Benn Torres et al., 2013*) show lower fractional Native American genetic ancestry.

During our fieldwork, it was noted that members of the Kalinago community characterized themselves and others in terms of perceived genealogical ancestry as 'Black,' 'Kalinago,' or 'Mixed.' Compared to individuals self-identified as 'Mixed,' those self-identified as 'Kalinago' have on average more Native American genetic ancestry (67% vs 51%), less European genetic ancestry (10% vs 14%), and less African genetic ancestry (23% vs 34%) (*Figure 2—figure supplement 2*). Thus, these folk categories based on phenotype are reflected in some underlying differences in genetic ancestry.

## Kalinago skin color variation

Melanin index unit (MI) calculated from skin reflectance measured at the inner upper arm (see Materials and methods) was used as a quantitative measure of melanin pigmentation (*Ang et al., 2012*; *Diffey et al., 1984*). MI determined in this way is commonly used as a measure of constitutive skin pigmentation (*Choe et al., 2006*; *Park and Lee, 2005*). The MI in the Kalinago ranged from 20.7 to 79.7 (*Figure 4—figure supplement 1*), averaging 45.7. The three Kalinago albino individuals sampled had the lowest values (20.7, 22.4, and 23.8). Excluding these, the MI ranged between 28.7 and 79.7 and averaged 45.9. For comparison, the MI averaged 25 and 21 for people of East Asian and European genetic ancestry, respectively, as measured with the same equipment in our laboratory (*Ang et al., 2012*; *Tsetskhladze et al., 2012*). This range is similar to that of another indigenous population, the Senoi of Peninsular Malaysia (MI 24–78; mean = 45.7) (*Ang et al., 2012*). The Senoi are believed to include admixture from Malaysian Negritos whose pigmentation is darker (mean = 55) (*Ang et al., 2012*) than that of the average Kalinago. In comparison, the average MI was 53.4 for Africans in Cape Verde (*Beleza et al., 2012*) and 59 for African-Americans (*Shriver et al., 2003*). Individuals self-described as 'Kalinago' were slightly lighter and had a narrower MI distribution (42.5± 5.6, mean ± SD) compared to 'Mixed' (45.8± 9.6) (*Figure 4—figure supplement 2*).

## An OCA2 albinism allele in the Kalinago

OCA is a genetically determined condition characterized by nystagmus, reduced visual acuity, foveal hypoplasia, and strabismus as well as hypopigmentation of the skin, hair, and eye (*Dessinioti et al., 2009*; *van Geel et al., 2013*). The three sampled albino individuals had pale skin (MI 20.7, 22.4, and 23.8 vs. 29–80 for non-albino individuals), showed nystagmus, and reported photophobia and high susceptibility to sunburn. In contrast to the brown irides and black hair of most Kalinago, including their parents, the albino individuals had blonde hair and gray irides with varying amounts of green and blue.

To identify the albinism variant in the Kalinago, we first determined that none of the albino individuals carried any of 28 mutations previously found in African or Native American albino individuals (*Carrasco et al., 2009*; *King et al., 2003*; *Stevens et al., 1997*; *Yi et al., 2003*), including a 2.7 kb exon 7 deletion in *OCA2* found at high frequency in some African populations. Whole exome sequencing of one albino individual and one parent (obligate carrier) revealed polymorphisms homozygous in the albino individuals and heterozygous in the parent, an initial approach that assumes that the albino individual was not a compound heterozygote. We identified 12 variant alleles in 7 OCA genes (or genomic regions) that met these criteria (summarized in *Appendix 1—table 4*). None were nonsense or splice site variants. Five of the twelve variants were intronic, one was synonymous, one was located in 5'UTR, and three were in the 3'UTR (*Appendix 1—table 4*). Two missense variants were found in *OCA2*: SNP rs1800401 (c.913C>T or p.Arg305Trp in exon 9), *R305W*, and multi-nucleotide polymorphism rs797044784 in exon 8 (c.819_822delCTGGinsGGTC; p.Asn273_Trp274delinsLysVal), *NW273KV*.

Among 458 Kalinago *OCA2* genotypes, 26 carried *NW273KV* and 60 carried *R305W* (*Table 1*). Only *NW273KV* homozygotes were albino individual. We know that the allele responsible for albinism was *NW273KV* because neither of the two individuals, homozygous for *R305W* but not *NW273KV*, was albino individual. In further support of this conclusion is that one individual who was homozygous for

**Table 1.** Albinism among *NW273KV* and *R305W* genotypes.

| Allele/genotype | | NW273KV genotype | | | |
| --- | --- | --- | --- | --- | --- |
| | | Homozygous ancestral* | Heterozygous | Homozygous derived | Total |
| *R305W* genotype | Homozygous ancestral | 398 | 0 | 0 | 398 |
| | Heterozygous | 33 | 22 | 0 | 55 |
| | Homozygous derived | 1 | 1 | 3† | 5 |
| | Total | 432 | 23 | 3† | 458 |

*Ancestral = reference allele and derived = alternate allele for both variants.
†Albino phenotype. Notably, none of the other genotypic categories are albino individuals.

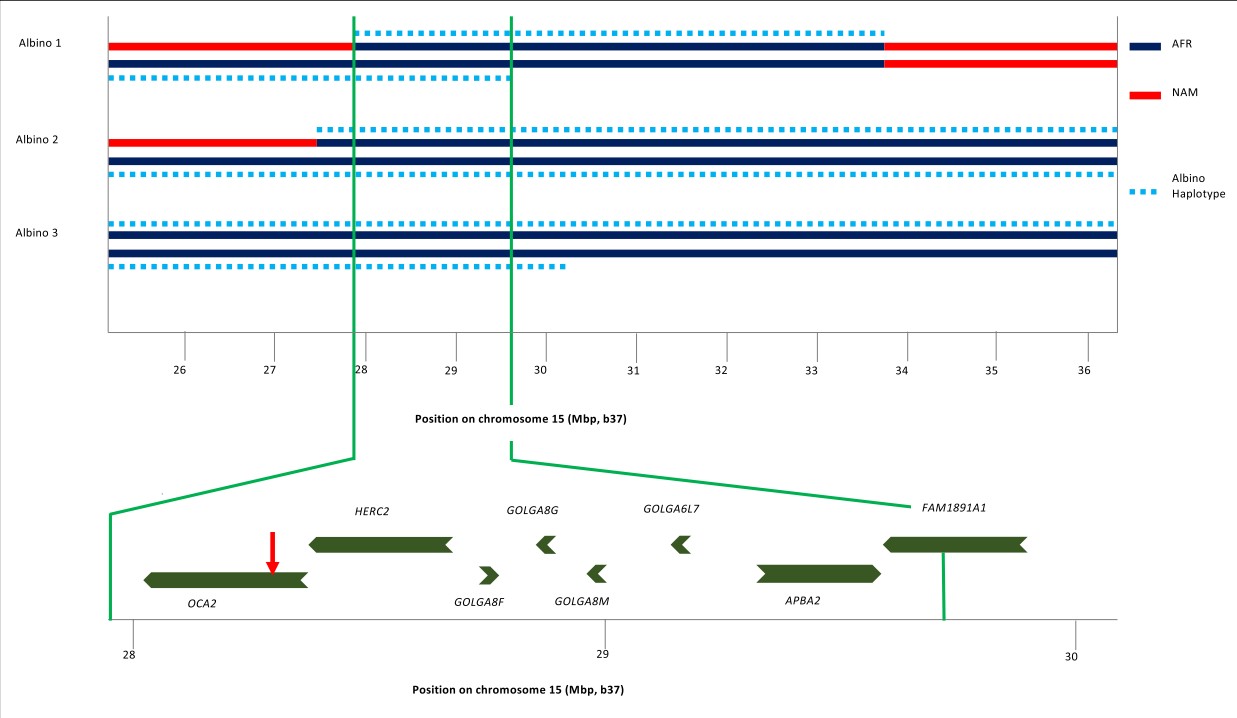

**Figure 3.** Haplotype analysis for three albino individuals. The inner two lines indicate NAM (red) or AFR (dark blue) genetic ancestry; no EUR genetic ancestry was found in this genomic region. For this local genetic ancestry analysis, the region shown here consisted of 110 non-overlapping segments with 7–346 SNPs each (mean 65). The deduced extent of shared albino haplotype (dotted light blue lines) is indicated on each chromosome. The common region of overlap indicated by the minimum homozygous region (determined by albino individual 1) shared by all three albino individuals is shown at expanded scale below. Genes in this region are labeled, and the position of the *NW273KV* polymorphism in *OCA2* is indicated by the red arrowhead.

*R305W* and homozygous ancestral for *NW273* had an MI of 72, among the darkest in the entire population. *R305W* is notably present with frequency >0.10 in some African, South Asian, and European populations (*Auton et al., 2015*), predicting a Hardy-Weinberg frequency of homozygotes above 1%. This is far greater than the observed frequency of individuals with albinism and therefore inconsistent with the idea that this is not a variant responsible for albinism. The fact that *R305W* scores incorrectly as pathogenic using SIFT, Polyphen 2.0, and PANTHER that *R305W* (*Kamaraj and Purohit, 2014*) suggests a need for refinement of these methods. The universal association of *R305W* with the *NW273KV* haplotype indicates that the founder haplotype of the *NW273KV* albinism mutation carried the silent *R305W* variant.

To identify the origin of the albino allele, albino individuals and carriers were analyzed for regions exhibiting homozygosity, and identity-by-descent and local genetic ancestry was estimated (see Materials and methods). All three albino individuals share a homozygous segment of ~1.7 Mb that encompasses several genes in addition to OCA2 (*Figure 3*). The albino haplotype defined by homozygosity in individuals 2 and 3 extends ~11 Mb; comparison to local genetic ancestry shows that this haplotype is clearly of African origin.

The Kalinago albino individuals are the only reported individuals where the albinism was caused by homozygosity for the *NW273KV* allele of *OCA2*. Two reported albino individuals of African-American/Dutch descent were compound heterozygotes for the *OCA2* mutation, with one allele being the *NW273KV* variant chromosome (*Garrison et al., 2004*; *Lee et al., 1994*). Conservation of the NW sequence among vertebrates and its inclusion in a potential N-linked glycosylation site (*Rinchik et al., 1993*) that is eliminated by the mutation supports the variant's pathogenicity. The *NW273KV* frequency in our sample (0.03) translates into a Hardy-Weinberg albinism frequency ($p^2$=0.0009) of ~1 per 1000, as observed (3 in a population of about 3000). Examination of publicly available data reveals three *OCA2*[NW273KV] heterozygotes in the 1000 Genome Project, a pair of siblings from Barbados (ACB) and one individual from Sierra Leone (MSL). The three 1KGP individuals share a haplotype of ~1.5 Mb,

of which ~1.0 Mb matches the albino haplotype in the Kalinago. The phasing for the $OCA2^{NW273KV}$ variant in the public data is inconsistent, with the variant assigned to the wrong chromosome for the ACB siblings.

## Genetic contributions to Kalinago skin color variation

One motivation for undertaking this work was to characterize genetic contributions to skin pigmentation in a population with primarily Native American and African genetic ancestry, so that we could focus on the effect of Native American hypopigmenting alleles without interference from European alleles. The Kalinago population described here comprises the only population we are aware of that fits this genetic ancestry profile. To control for the effects of the major European pigmentation loci, all Kalinago samples were genotyped for $SLC24A5^{A111T}$ and $SLC45A2^{L374F}$. The phenotypic effects of these variants and $OCA2^{NW273KV}$ are shown in *Figure 4*. Each variant decreases melanin pigmentation, with homozygotes being lighter than heterozygotes. The greatest effect is seen in the $OCA2^{NW273KV}$ homozygotes (the albino individuals), as previously noted. The frequencies of the derived alleles of $SLC24A5^{A111T}$ and $SLC45A2^{L374F}$ in the Kalinago sample are 0.14 and 0.06, respectively.

The markedly higher frequency of $SLC24A5^{A111T}$ compared to $SLC45A2^{L374F}$ is not explained solely by European admixture, given that most Europeans are nearly fixed for both alleles (*Soejima and Koda, 2007*). This deviation can be explained by the involvement of source populations that carry the $SLC24A5^{A111T}$ variant but not $SLC45A2^{L374F}$. Although some sub-Saharan West African populations (the likeliest source of AFR genetic ancestry in the Kalinago) have negligible $SLC24A5^{A111T}$ frequencies, moderate frequencies are found in the Mende of Sierra Leone (MSL, allele frequency = 0.08) (*Micheletti et al., 2020*; *Auton et al., 2015*), while some West African populations such as Hausa and Mandinka who have allele frequencies of 0.11 and 0.15, respectively (*Cheung et al., 2000*; *Rajeevan et al., 2012*). Such African individuals carrying the $SLC24A5^{A111T}$ allele could potentially cause the observed frequencies by founder effect. In addition, the region of chromosome 5 containing $SLC45A2$ exhibits low European genetic ancestry (6.5%) that is consistent with low observed $SLC45A2^{L374F}$ frequency.

In order to investigate the potential effect of the $SLC25A5^{A111T}$ allele on the albinism phenotype, we also compared other pigmentation phenotypes such as the hair and eye colors for all albino individuals and carriers. One of the three Kalinago albino individuals was also heterozygous for $SLC24A5^{A111T}$, but neither skin nor hair color for this individual was lighter than that of the other two albino individuals, who were homozygous for the ancestral allele at $SLC24A5^{A111}$; this observation is consistent with epistasis of $OCA2$ hypopigmentation over that of $SLC24A5^{A111T}$. Nine sampled non-albino individuals had combinations of hair that was reddish, yellowish, or blonde (n=6), skin with MI <30 (n=3), and gray, blue, green, or hazel irides (n=2); among these, six were heterozygous and one homozygous for $SLC24A5^{A111T}$, and three were heterozygous for the albino variant. A precise understanding of the phenotypic effects of the combinations of these and other hypopigmenting alleles will require further study.

The strong dependence of pigmentation on Native American genetic ancestry is clarified by focusing on individuals lacking the hypopigmenting alleles $SLC24A5^{A111T}$, $SLC45A2^{L374F}$, and $OCA2^{NW273KV}$ (*Figure 5*). Although positive deviations from the best fit are apparent at both high and low Native American genetic ancestry, the trend toward lighter pigmentation as Native American genetic ancestry increases is clear. The net difference between African and Native American contributions to pigmentation appears likely to be bounded by the magnitudes of the slope vs NAM genetic ancestry (24 units) and the slope vs AFR genetic ancestry (29 units, not shown). The difference in melanin index value is expected to be explained by genetic variants that are highly differentiated between African and Native American populations.

To further investigate the contributions of genetic variation to skin color, we performed association analyses using an additive model for melanin index, conditioning on sex, genetic ancestry (using 10 PCs), and genotypes for $SLC24A5^{A111T}$, $SLC45A2^{L374F}$, and $OCA2^{NW273KV}$. Assuming likely epistasis of albinism alleles over other hypopigmenting alleles, these analyses omitted the three albino individuals. Employing a linear regression model, we found that sex and all three genotyped polymorphisms were statistically significant (*Table 2* and *Figure 2—figure supplement 2*). However, only $SLC24A5^{A111T}$ reaches genome-wide significance. PC1, which strongly correlated with Native American

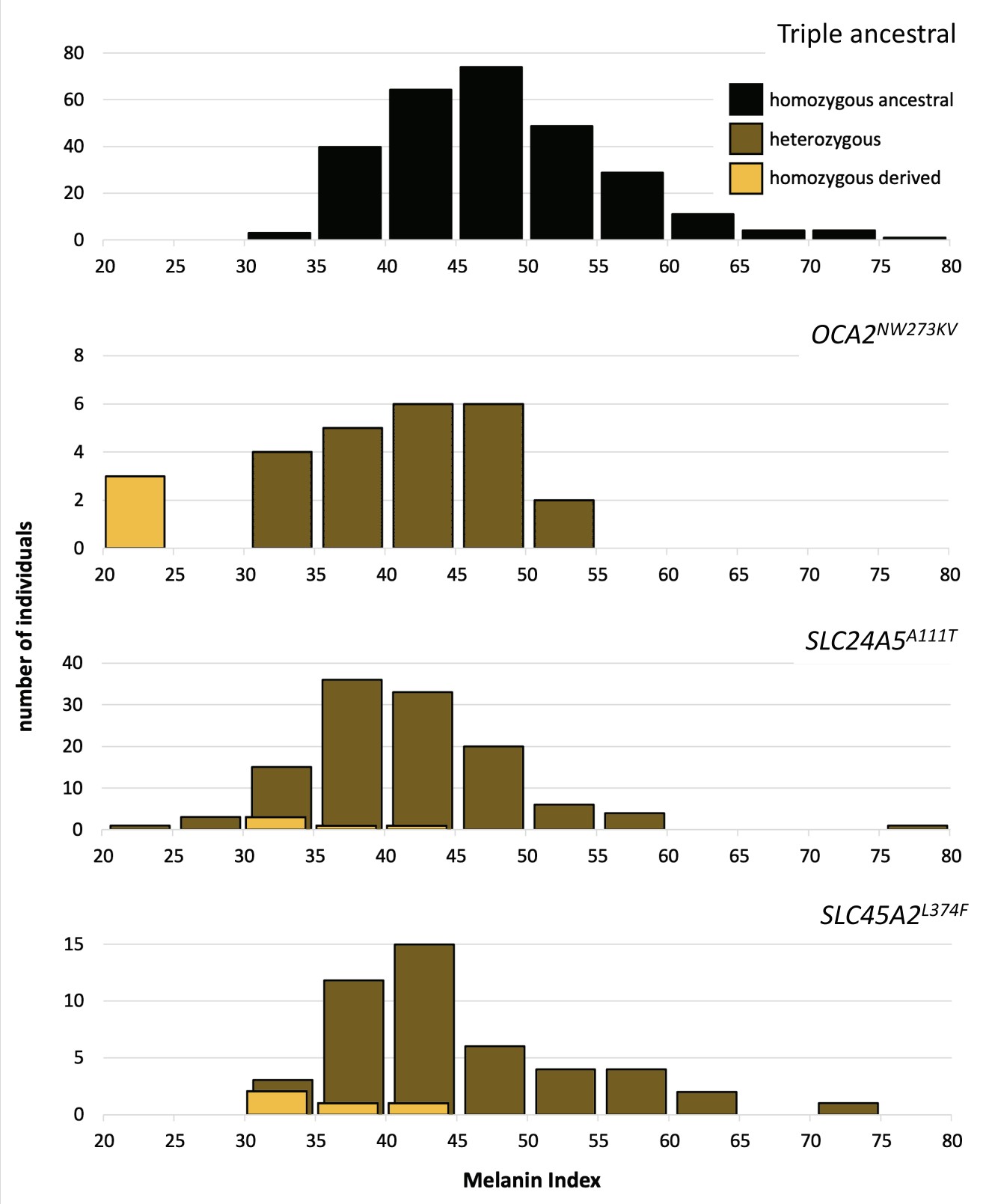

**Figure 4.** Skin color distribution of Kalinago samples according to genotype. The 'triple ancestral' plot is individuals ancestral for three pigmentation loci (*SLC24A5*[111A], *SLC45A2*[374L], and *OCA2*[273NW]). In the other plots, heterozygosity or homozygosity is indicated for the variants: *OCA2*[NW273KV]; *SLC24A5*[A111T]; and *SLC45A2*[L374F]. Individuals depicted in the second through fourth panels are repeated if they carry variants at more than one locus. M-

*Figure 4 continued on next page*

*Figure 4 continued*

index of the Kalinago ranged from 20.7 to 79.7 (***Figure 4—figure supplement 1***) and the histogram of skin color based on community-defined ancestry are shown in ***Figure 4—figure supplement 2***.

The online version of this article includes the following source data and figure supplement(s) for figure 4:

**Source data 1.** The source file contain melanin index distribution as function of community-described ancestry.

**Source data 2.** The source data contains data of melanin indices according to genotype.

**Figure supplement 1.** Skin color distribution of the Kalinago from Commonwealth of Dominica.

**Figure supplement 2.** Melanin index distribution as function of community-described ancestry.

vs African genetic ancestry, exhibits the lowest p-value. Effect sizes were about –6 units (per allele) for *SLC24A5*$^{A111T}$, –4 units for *SLC45A2*$^{L374F}$, and –8 units for the first *OCA2*$^{NW273KV}$ allele.

Additional covariates were considered but not included in our standard model. Skin pigmentation exhibited a decreasing trend with age, but its contribution was not statistically significant (adjusted p-value = 0.08). Estimated effect sizes for significant covariates were little affected by the inclusion of age as a covariate (***Appendix 1—table 5***). Analysis of SNPs that were previously reported as relevant to pigmentation are shown in ***Appendix 2—table 1***. The lowest (adjusted) p-value for this collection

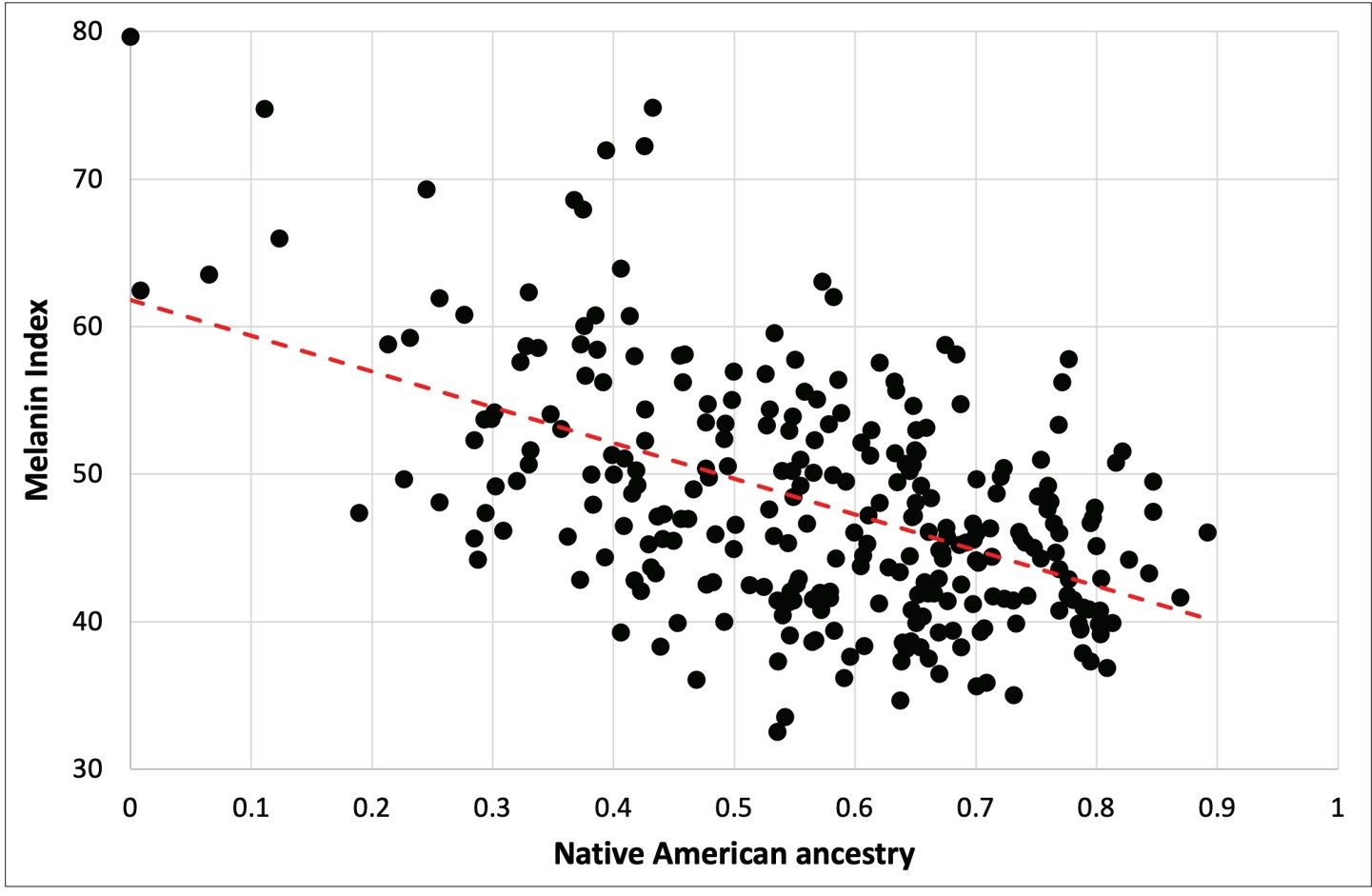

**Figure 5.** Dependence of melanin unit on genetic ancestry for Kalinago. Only individuals who are ancestral for *SLC24A5*$^{111A}$, *SLC45A2*$^{374L}$, and *OCA2*$^{273NW}$ alleles are shown (n=279). The dotted red line represents the best fit (linear regression). Slope is –24.3 (melanin index unit [MI] = –24.3*NAM+61.9); r²=0.2722.

The online version of this article includes the following figure supplement(s) for figure 5:

**Figure supplement 1.** Estimated power for GWAS using Kalinago sample.

**Figure supplement 2.** Q-Q plots for association analyses to identify novel SNPs that may contribute towards skin pigmentation in the Kalinago samples.

**Table 2.** Effect sizes for covariates in linear regression model with 10 principal components.

| Covariate | Effect size (MI) | p-Value |
| --- | --- | --- |
| rs1426654 ($SLC24A5^{A111T}$) | −5.8 | 1.5E-12 |
| rs16891982 ($SLC45A2^{L374F}$) | −4.4 | 6.7E-05 |
| Albino allele ($OCA2^{NW273KV}$) | −7.7 | 2.2E-05 |
| Sex (female vs male) | −2.4 | 5.0E-04 |

[a]Per allele effect size, in melanin units, for A111T and L374F; effect of first allele for albino variant.

of variants is about 0.001, considerably larger than the p-values for the variants included as covariates in our standard model. Inclusion of the SNP of lowest p-value from each of the five regions containing *BCN2*, *TYR*, *OCA2*, *MC1R*, and *OPRM1* only modestly altered effect sizes for the other covariates (*Appendix 1—table 5*).

The effect size for $SLC24A5^{A111T}$ measured here is consistent with previously reported results of −5 melanin units calculated from an African-American sample (*Lamason et al., 2005*; *Norton et al., 2007*) and −5.5 from admixed inhabitants of the Cape Verde islands (*Beleza et al., 2013*). Reported effect sizes for continental Africans are both higher and lower, −7.7 in *Crawford et al., 2017*, and −3.6 *Martin et al., 2017b*, while the estimated effect size in the CANDELA study (GWAS of combined admixed populations from Mexico, Brazil, Columbia, Chile, and Peru) (*Adhikari et al., 2019*) yielded an effect size about −3 melanin units.

A significant effect of $SLC45A2^{L374F}$ on skin pigmentation reported for the African-American sample by *Norton et al., 2007*, and in the CANDELA study by *Adhikari et al., 2019*, but not for the African Caribbean sample by *Norton et al., 2007*. The 4 unit effect size of this allele in the Kalinago reported here is similar to the 5 unit effect reported by *Norton et al., 2007*. *Beleza et al., 2013* reported significance for an SNP in strong linkage disequilibrium with $SLC45A2^{L374F}$, which was itself not genotyped.

Our estimate that a single $OCA2^{NW273KV}$ allele causes about −8 melanin units of skin lightening is the first reported population-based effect size measurement for any albinism allele. Although albinism is generally considered recessive, our population sample offered an opportunity to compare the effect size for the first and second alleles quantitatively. We applied the estimated parameters to the three albino individuals and found that they were lighter by an average of 10 uni nm, *05W* homozygotes, when controlling for $OCA2^{NW273KV}$ status, $OCA2^{R305W}$ had no detectable effect on skin color (*Appendix 2—table 1*).

To identify novel SNPs that may contribute toward skin pigmentation in the Kalinago samples, we performed GWAS using linear regression and linear mixed models (LMMs). Estimated power for these analyses is shown in *Figure 5—figure supplement 1*, and Q-Q plots are depicted in *Figure 5—figure supplement 2*. The LMM approaches exhibited less statistic inflation than linear regression, likely because they better accounted for closely related individuals. Although the lowest p-values from the LMM-based methods meet the conventional criterion of 5e-08 for genome-wide significance (*Appendix 3—table 1*), our interpretation is that none of these variants warrant further investigation. Low observed minor allele frequencies (<2%) are inconsistent with those expected for variants responsible for pigmentation differences between the African and Native American populations because the frequencies of alleles responsible for population differences are expected to be highly differentiated between these source populations.

Additional Native American hypopigmenting alleles of significant effect size remain to be identified. Previously characterized variants do not explain this difference. It is possible that multiple hypopigmenting variants of small effect sizes are together required to reach Native American and/or East Asian levels of hypopigmentation, individually having insufficient effect to detect in the Kalinago, given our power limitations. If this is the case, multiple variants are required to explain the observed net difference in pigmentation. Alternatively, if there are variants with large effect sizes, it appears likely that they were not genotyped and are poorly tagged by the genotyped SNPs. Additional work will be required to find hypopigmentation alleles of significant effect size that are responsible for the lighter color of Native Americans.

# Materials and methods

## Recruitment

Participants from among the Kalinago populations were recruited with the help of nurses from the Kalinago Territory in 2014. Recruitment took place throughout the territory's eight hamlets. Place and date of birth, reported genealogical ancestry of parents and grandparents, number of siblings, and response to sun exposure (tanning ability, burning susceptibility) were obtained by interview. Hair color and texture and eye color (characterized as black, brown, gray, blue, green, hazel, no pigment) were noted visually but not measured quantitatively.

## Skin reflectometry

Skin reflectance was measured using a Datacolor CHECK[PLUS] spectrophotometer and converted to melanin unit as we have previously described (*Ang et al., 2012*; *Diffey et al., 1984*). To minimize the confounding effects of sun exposure and body hair, skin color measurements were measured on each participant's inner arm, and the average of triplicate measurements was generated. Before skin color measurements were taken, alcohol wipes were used to minimize the effect of dirt and/or oil. In order to minimize blanching due to occlusion of blood from the region being measured, care was taken not to apply only sufficient pressure to the skin to prevent ambient light from entering the scanned area (*Fullerton et al., 1996*).

## DNA collection

Saliva samples were collected using the Oragene Saliva kit, and DNA was extracted using the prepIT. L2P kit, both from DNA Genotek (Ottawa, Canada). DNA integrity was checked by agarose gel electrophoresis and quantitated using a NanoDrop spectrophotometer (Thermo Fisher Scientific, Waltham, MA, USA). Further quantification was done using Qubit Fluorometer (Thermo Fisher Scientific, Waltham, MA, USA) as needed, following the manufacturer's instructions.

## Genotyping

OCA variants previously identified in African and Native Americans (*Carrasco et al., 2009*; *King et al., 2003*; *Stevens et al., 1997*; *Yi et al., 2003*) were amplified by PCR in all albino individuals as well as control samples using published conditions. Selected alleles of *SLC24A5, SLC45A2, OCA2,* and *MFSD12* were amplified in all sampled individuals as described in *Appendix 1—table 6*. Amplicons generated by 30 cycles of PCR using an Eppendorf thermocycler were sequenced (GeneWiz, South Plainfield, NJ, USA) and the chromatograms viewed using Geneious software.

Illumina SNP genotyping using the Infinium Omni2.5–8 BeadChip was performed for all the individuals sampled. This was performed in three cohorts, using slightly different versions of the array, and the results combined. Due to ascertainment differences between the cohorts, analysis is presented here only for the combined sample. After quality control to eliminate duplicates and monomorphic variants, and to remove variants and individuals with genotype failure rates >0.05, 358 Kalinago individuals and 1,638,140 unique autosomal SNPs remained.

## Whole exome sequencing of albino individual and obligate carrier

In order to identify the causative variant for albinism in the Kalinago, two samples (one albino individual and one parent) were selected for whole exome sequencing. Following shearing of input DNA (1 µg) using a Covaris E220 Focused-ultrasonicator (Woburn, MA, USA), exome enrichment and library preparation was done using the Agilent SureSelect V5+UTR kit (Santa Clara, CA, USA). The samples were sequenced at 50× coverage using a HiSeq 2500 sequencer (Illumina, San Diego, CA, USA).

The *fastq* files were aligned back to Human Reference Genome GRCh37 (HG19) using BWA (*Li and Durbin, 2009*) and bowtie (*Langmead et al., 2009*). Candidate SNP polymorphisms were identified using GATK's UnifiedGenotyper (*McKenna et al., 2010*), while the IGV browser was used to examine the exons of interest for indels (*Thorvaldsdóttir et al., 2013*). Variants with low sequence depth (<10) in either sample were excluded from further consideration.

## Computational analysis

Basic statistics, merges with other datasets, and association analysis by linear regression were performed using plink 1.9 (*Chang et al., 2015*; *Purcell et al., 2007*). Phasing and imputation, as well as analysis of regions of homozygosity by descent and identity by descent were performed with Beagle 4.1 (*Browning and Browning, 2013*; *Browning and Browning, 2007*), using 1KGP phased data (*Auton et al., 2015*) as reference.

The genotyped individuals were randomly partitioned into nine subsets of 50 or 51 individuals (n=50 subsets) in which no pair exhibited greater than second-order relationship (PI_HAT >0.25 using the `--genome` command in plink). Using the same criteria, a maximal subset of 184 individuals was also generated (n=184 subset).

PCA was performed using the smartpca program (version 13050) in the eigensoft package (*Price et al., 2006*). For comparison to HGDP populations, Kalinago samples were projected onto PCs calculated for the HGDP samples alone. For use as covariates in association analyses, the n=184 subset was used to generate the PCA, and the remaining individuals were projected onto the same axes.

Admixture analysis was performed using the ADMIXTURE program (*Alexander et al., 2009*; *Zhou et al., 2011*). Each of the nine n=50 Kalinago subsets was merged with the N=940 subset of HGDP data (*Li et al., 2008*; *Rosenberg, 2006*) for analysis (349,923 SNPs) and the outputs combined, averaging genetic ancestry proportions for the common HGDP individuals across runs. These results were used in figures. Separately, two-stage admixture analysis started with the averaged estimated allele frequencies and then employed the projection (--P) matrix outputs to estimate individual genetic ancestry for the combined Kalinago sample. Individual ancestries estimated using both methods, as well as those estimated from a thinned subset of 50,074 SNPs were in good agreement, consistent with standard errors estimated by bootstrap analysis, although sample-wide averages differed slightly. Cross-validation is enabled by adding the `--cv` to the ADMIXTURE command.

For association analyses we removed the three-albino individuals and excluded SNPs with minor allele frequency <0.01. For conventional association analysis by linear regression, the standard additive genetic model included sex, the first 10 PCs, and genotypes of rs1426654 (*SLC24A5*), rs16891982 (*SLC45A2*), and the albino variant rs797044784 (*OCA2*) as covariates (*Supplementary file 4*). LMM analysis was performed using the mlma module of GCTA (*Yang et al., 2011*) with the `--mlma-no-preadj-covar` flag to suppress calculation using residuals. Two genetic relatedness matrices (GRM) were used: a standard GRM calculated using GCTA's --make-grm command and an ancestry-aware GRM calculated using relationships deduced by REAP (*Thornton et al., 2012*) that utilized the output of the two-stage admixture analysis. For linear regression only, p-values were adjusted for statistic inflation by genomic control using the lambda calculated from the median chi-square statistic.

Statistical power was estimated by simulation, using a subset of genotyped SNPs. Starting with the 349,923 SNPs used for genetic ancestry analysis, the averaged P matrix from ADMIXTURE analysis at K=4 provided an initial estimate of allele frequencies in AFR and NAM ancestral populations; 10,233 SNPs exhibited differentiation of 0.7 or greater between these populations, a value chosen as a reasonable minimum population differentiation for causative variants. After removal of SNPs for which predicted Kalinago sample frequencies deviated by more than 0.1 from observed values and those with adjusted p<0.1, 8766 SNPs remained. Phenotypes were simulated by randomly selecting one of these SNPs and adding a defined effect size to the observed phenotype. Simulated datasets were then analyzed with plink using the standard genetic model.

Statistical analysis of pigmentary effect of albinism involved fitting parameters to an additive model for the sample containing carriers but lacking albino individuals, applying the same model to the albino individuals, and comparing residuals for the albinos and the other individuals.

Local genetic ancestry analysis of the region containing the albinism allele was performed using the PopPhased version of rfmix (v1.5.4) with the default window size of 0.2 cM (*Maples et al., 2013*). A subset of 1KGP data served as reference haplotypes for European, African, and East Asian populations, and the Native American genetic ancestry segments of the admixed samples as determined by *Martin et al., 2017a*, were combined to generate synthetic Native American reference haplotypes. For estimates of individual genetic ancestry, Viterbi outputs for each window were averaged across all autosomes.

## Acknowledgements

We would like to thank the Kalinago Council, Dominica Ministry of Health, nurses at the Kalinago Territory, Salybia Mission Project, and the Kalinago community for their assistance and participation in this study. We would also like to acknowledge faculty of Ross University, Portsmouth, Dominica (now Bridgetown, Barbados), especially Drs. Gerhard Meisenberg (retired) and Liris Benjamin of Ross University in helping us to obtain the necessary IRB approval for fieldwork. This work was supported by the Hershey Rotary Club, Microryza (now Experiment.com), Jake Gittlen Laboratories for Cancer Research, National Institutes of Health grants 5R01 AR052535 and 3R01 AR052535-03S1 from the National Institute of Arthritis and Musculoskeletal and Skin Diseases, and Department of Pathology for funding portions of this project. We would also like to acknowledge members of the Cheng Lab for their constructive comments and input.

## Additional information

### Funding

| Funder | Grant reference number | Author |
|---|---|---|
| Hershey Rotary Club | | Khai C Ang |
| Jake Gittlen Laboratories for Cancer Research | | Keith C Cheng |
| Department of Pathology, Penn State College of Medicine | | Keith C Cheng |
| Microryza (now Experiment.com) | | Khai C Ang |
| National Institute of Arthritis and Musculoskeletal and Skin Diseases | 5R01 AR052535 | Keith C Cheng |
| National Institute of Arthritis and Musculoskeletal and Skin Diseases | 3R01 AR052535-03S1 | Keith C Cheng |

The funders had no role in study design, data collection, and interpretation, or the decision to submit the work for publication.

### Author contributions

Khai C Ang, Conceptualization, Resources, Data curation, Software, Formal analysis, Supervision, Funding acquisition, Validation, Investigation, Visualization, Methodology, Writing - original draft, Project administration, Writing - review and editing; Victor A Canfield, Formal analysis, Validation, Investigation, Visualization, Methodology, Writing - review and editing; Tiffany C Foster, Data curation, Investigation, Methodology, Writing - original draft; Thaddeus D Harbaugh, Kathryn A Early, Rachel L Harter, Katherine P Reid, John W Hawley, Investigation; Shou Ling Leong, Conceptualization; Yuka Kawasawa, Dajiang Liu, Formal analysis; Keith C Cheng, Conceptualization, Resources, Supervision, Funding acquisition, Investigation, Visualization, Methodology, Writing - review and editing

### Author ORCIDs

Khai C Ang http://orcid.org/0000-0001-7695-9953
Yuka Kawasawa http://orcid.org/0000-0002-8638-6738
Keith C Cheng http://orcid.org/0000-0002-5350-5825

### Ethics

Human subjects: The study was reviewed and approved by the Kalinago council and institutional review boards of Penn State University (29269EP), Ross University, and the Dominica Ministry of Health (H125). Informed consent was obtained from each participant enrolled in the study, and in the case of minors, consent was also obtained from a parent or guardian.

**Decision letter and Author response**
Decision letter https://doi.org/10.7554/eLife.77514.sa1
Author response https://doi.org/10.7554/eLife.77514.sa2

## Additional files

### Supplementary files

- MDAR checklist
- Supplementary file 1. Kalinago Sample Demographics.
- Supplementary file 2. Effect sizes of previously reported variants in Kalinago samples.
- Supplementary file 3. Top novel variants that may contribute towards skin pigmentation from our GWAS analysis.
- Supplementary file 4. Table S9: PCA data with sex and genotypes as covariates.

### Data availability

The whole exome sequencing and whole genome SNP genotyping data underlying this article cannot be shared publicly due to the privacy of individuals and stipulation by the Kalinago community. Only de-identified filtered SNP data used in analyses will be shared. Additional data will be shared on request to the corresponding author, pending approval from the Kalinago Council. M-index and specific genotyping data (SLC24A5 A111T, SLC45A2 L374F, OCA2 NW273KV and OCA2 305W) and genotyping data for Admixture have been uploaded to Dryad https://doi.org/10.5061/dryad. sf7m0cg7z. The data cannot be used for any commercial purposes. We did not create any new software or script for analysis.

The following dataset was generated:

| Author(s) | Year | Dataset title | Dataset URL | Database and Identifier |
|---|---|---|---|---|
| Ang KC, Canfield VA, Foster TC, Harbaugh TD, Early KA, Harter R, Harbaugh T, Early K, Harter R, Reid KP, Leong S, Imamura Kawasawa Y, Liu D, Hawley JW, Cheng KC | 2023 | Native American Genetic Ancestry and Pigmentation Allele Contributions to Skin Color in a Caribbean Population | https://dx.doi.org/10.5061/dryad.sf7m0cg7z | Dryad Digital Repository, 10.5061/dryad.sf7m0cg7z |

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

## Appendix 1

### Supplementary Tables

**Appendix 1—table 1.** Sample Demographics.

| Category | Entire sample (N=461) |
| --- | --- |
| **Sex** | |
| male | 244 |
| female | 217 |
| **Age** | |
| range | 6–93 |
| mean (SD) | 39 (21.5) |
| median | 39 |
| **Paternal ancestry** | |
| reported* | 432 |
| named | 193 |
| sampled† | 49 |
| **Maternal ancestry** | |
| reported* | 437 |
| named | 244 |
| sampled† | 128 |

*community-described ancestry collected.
†values from reported genealogy; 75 fathers and 146 mothers as determined by genotyping.

**Appendix 1—table 2.** Summary of Kalinago ancestry from admixture analysis (n=458).
NAM = Native American, AFR = African, EUR = European, CSA = Central & South Asian, EAS = East Asian, OCE = Oceanian. At K=3, NAM, EAS, and OCE are not distinguishable.

| K-value | AFR | NAM | EAS | OCE | EUR | CSA |
| --- | --- | --- | --- | --- | --- | --- |
| 3 | 0.304 | 0.552 | | | 0.144 | |
| 4 | 0.318 | 0.549 | 0.011 | | 0.122 | |
| 5 | 0.318 | 0.548 | 0.011 | 0.002 | 0.121 | |
| 6 | 0.318 | 0.548 | 0.012 | 0.002 | 0.110 | 0.010 |

**Appendix 1—table 3.** Ancestry proportions estimated using different approaches.

| estimation approach | AMR | AFR | EUR | EAS |
| --- | --- | --- | --- | --- |
| Admixture (subsets, K=4) | 0.549 | 0.318 | 0.122 | 0.011 |
| Admixture (two stage, K=4) | 0.541 | 0.316 | 0.126 | 0.016 |
| rfmix (4 clusters) | 0.553 | 0.313 | 0.125 | 0.009 |
| rfmix (3 clusters) | 0.557 | 0.326 | 0.117 | --- |

**Appendix 1—table 4.** Summary by locus of albinism candidates identified through exome sequencing.
Candidates are homozygous derived in one albino and heterozygous in one obligate carrier. No nonsense, frameshift, or splice variants was detected. Our initial attempt to identify the albinism variant in the Kalinago involved targeted genotyping of the albino individuals for 28 mutations previously observed (**Honychurch, 2012**; **Honychurch, 1998**; **Li et al., 2008**; **Loomis, 1967**) in

African or Native American albinos; these included the 2.7 kb exon 7 deletion in *OCA2* found at high frequency in some African populations. No mutation was detected using this approach.

**A.**

| OCA gene | Chromosome | Variants | Missense |
|---|---|---|---|
| OCA1 (TYR) | 11 | 0 | |
| OCA2 | 15 | 5 | 2 |
| OCA3 (TYRP1) | 9 | 0 | |
| OCA4 (SLC45A2) | 5 | 0 | |
| OCA5 | 4 | 6 | 0 |
| OCA6 (SLC24A5) | 15 | 0 | |
| OCA7 (LRMDA) | 10 | 1 | 0 |

**B. Characteristics of individual candidates identified through exome sequencing**

| Chr | rsID | Ref | Alt | f(AFR)* | Gene | Location/ Effect |
|---|---|---|---|---|---|---|
| 4 | rs3733437 | T | C | 0.126 | EMCN | intron |
| 4 | rs6826912 | T | G | 0.327 | PPP3CA | 3'UTR |
| 4 | rs463373 | T | C | 0.986 | SLC39A8 | 3'UTR |
| 4 | rs439757 | C | A | 0.986 | SLC39A8 | 3'UTR |
| 4 | rs223495 | A | G | 0.399 | MANBA | intron |
| 4 | rs3733632 | A | G | 0.819 | TACR3 | 5'UTR |
| 10 | rs7911113 | A | G | 0.476 | LRMDA | intron |
| 15 | rs1800419 | A | G | 0.629 | OCA2 | synonymous |
| 15 | rs1800401 | G | A | 0.126 | OCA2 | R305W |
| 15 | rs797044784[†] | CCAG | GACC | 0.002 | OCA2 | NW273KV |
| 15 | rs73375883 | G | A | 0.203 | OCA2 | intron |
| 15 | rs972334 | G | A | 0.217 | OCA2 | intron |

*Overall frequency for non-reference allele in seven 1KGP African populations.
[†]1KGP describes this variant as four consecutive SNPs rs549973474, rs569395077, rs538385900 and rs558126113.

**Appendix 1—table 5.** Effect sizes for covariates in linear regression model with 10 Principal Components. Effect sizes are per allele for genomic variants (first allele only for albino variant). PC1 variance for included individuals (n=452) is 0.0045. P values adjusted using genomic control (applied to GWAS on the full variant set) are omitted if raw P value is above 0.05.

| variable | Gene | standard | BETA | P_raw | Q (-log P) | alt1 | BETA | P_raw | Q | Beta-ratio-std |
|---|---|---|---|---|---|---|---|---|---|---|
| rs7118677 | GRM5/Tyr | test | -2.419 | 0.0001392 | 3.86 | <omitted> | | | | |
| rs1800404 | OCA2 | test | -1.497 | 0.001446 | 2.84 | <omitted> | | | | |
| rs885479 | MC1R | test | -1.31 | 0.008565 | 2.07 | <omitted> | | | | |
| rs6917661 | OPRM1 | test | -1.084 | 0.0117 | 1.93 | <omitted> | | | | |
| rs2153271 | BNC2 | test | -2.27 | 0.0002337 | 3.63 | <omitted> | | | | |
| NW273KV | OCA2 | covariate | -7.744 | 9.00E-07 | 6.05 | covariate | -7.613 | 1.28E-06 | 5.89 | 0.98 |
| A111T | SLC24A5 | covariate | -5.757 | 2.45E-16 | 15.61 | covariate | -5.695 | 4.17E-16 | 15.38 | 0.99 |
| L374F | SLC45A2 | covariate | -4.415 | 3.90E-06 | 5.41 | covariate | -4.392 | 4.06E-06 | 5.39 | 0.99 |
| PC1 | | covariate | -62.14 | 5.73E-32 | 31.24 | covariate | -59.46 | 3.58E-28 | 27.45 | 0.96 |
| PC2 | | covariate | -0.0571 | 0.9904 | 0.00 | covariate | 0.7866 | 0.869 | | |

*Appendix 1—table 5 Continued on next page*

*Appendix 1—table 5 Continued*

| variable | Gene | standard | BETA | P_raw | Q (-log P) | alt1 | BETA | P_raw | Q | Beta-ratio-std |
|----------|------|----------|------|-------|-----------|------|------|-------|---|----------------|
| PC3 | | covariate | 7.231 | 0.1208 | 0.92 | covariate | 7.891 | 0.09021 | | |
| PC4 | | covariate | -2.677 | 0.6025 | 0.22 | covariate | -2.242 | 0.6618 | | |
| PC5 | | covariate | -6.993 | 0.1447 | 0.84 | covariate | -6.714 | 0.1602 | | |
| PC6 | | covariate | 1.642 | 0.7392 | 0.13 | covariate | 1.309 | 0.79 | | |
| PC7 | | covariate | 7.277 | 0.1385 | 0.86 | covariate | 6.716 | 0.1707 | | |
| PC8 | | covariate | 3.267 | 0.5061 | 0.30 | covariate | 2.667 | 0.5867 | | |
| PC9 | | covariate | -17.27 | 0.001093 | 2.96 | covariate | -16.19 | 0.002222 | | |
| PC10 | | covariate | 8.599 | 0.116 | 0.94 | covariate | 8.423 | 0.1224 | | |
| SEX | | covariate | 2.547 | 5.49E-05 | 4.26 | covariate | 2.515 | 6.45E-05 | 4.19 | 0.99 |
| AGE | | <omitted> | | | | covariate | -0.02985 | 0.04596 | | |

This table compares three versions of analysis (linear regression only).

P values reported here (and Q = – log P) are not corrected for statistic inflation.

The last column for each non-standard case shows ratio of the effect size to that for the standard model, omitting PCs other than.

alt1 model adds age to standard analysis.

alt2 model adds five additional SNPs to standard analysis.

**Appendix 1—table 6.** Amplification conditions used for genotyping Kalinago samples for the selected alleles.

| Gene & Variant | Primer Sequence | PCR Annealing Temperature (°C) |
|----------------|-----------------|-------------------------------|
| SLC24A5$^{A111T}$ rs1426654 | Fwd- CTCACCTACAAGCCCTCTGC Rev- AATTGCAGATCCAAGGATGG | 55 |
| SLC45A2$^{L374F}$ rs16891982 | Fwd- CCTGCTGGGACTCATCCATC Rev- AGCAGAGTGCATGAGAAGGG | 55 |
| OCA2$^{NW273KV}$ rs797044784 | Fwd- AGAGTCCCAGATGGTGTCTCA Rev- AGGTCAGACTCCTTTAAACG | 53 |
| OCA2$^{R305W}$ rs1800401 | Fwd- AGAGGGAGGTCCCCTAACTG Rev- ATCTCAAGCCTCCCTGACTG | 53 |
| MFSD12$^{Y182H}$ rs2240751 | Fwd- CCCAGGTGGAATAGCAGTGAG Rev- AGTGGTTGGAATCACCTGTCA | 61 |

## Appendix 2

### Supplementary Tables

**Appendix 2—table 1.** Effect sizes of previously reported variants in Kalinago samples.

| CHR | pos (b37) | SNP | REF | ALT | gene | location | CADD_PHRED | Polyphen (main) | SIFT (main) | Freq | GT source | AR2 | BETA_a | P_a_raw | P_a_adj | BETA_b | P_b | BETA_c | P_c | BETA_d | P_d | reference(s) |
|---|---|---|---|---|---|---|---|---|---|---|---|---|---|---|---|---|---|---|---|---|---|---|
| 2 | 25329016 | rs12233134 | C | T | EFR3B near POMC | intronic | 0.348 | - | - | 0.473 | IMP | 1 | 0.61 | 0.197 | 0.2653 | -0.06 | 0.910365 | 0.15 | 0.771545 | 0.23 | 0.657335 | Quillen et al., 2012 |
| 6 | 457748 | rs4959270 | C | A | LOC105374875 near IRF4 | intronic | 1.041 | - | - | 0.329 | GT | 1 | 0.07 | 0.8796 | 0.8959 | -0.14 | 0.783747 | -0.08 | 0.880465 | -0.07 | 0.896826 | Sulem et al., 2007 |
| 6 | 466033 | rs1540771 | C | T | LOC105374875 near IRF4 | intronic | 0.95 | - | - | 0.305 | GT | 1 | -0.02 | 0.9703 | 0.9744 | -0.15 | 0.765123 | -0.08 | 0.875216 | -0.09 | 0.853243 | Sulem et al., 2007 |
| 6 | 154663568 | rs2333857 | A | G | IPCEF1 near OPRM1 | intronic or upstream | 3.27 | - | - | 0.813 | IMP | 1 | 1.33 | 0.02932 | 0.05976 | 0.78 | 0.225651 | 1.16 | 0.0719104 | 1.22 | 0.059139 | Quillen et al., 2012 |
| 6 | 154721557 | rs6917661 | C | T | CNKSR3 near OPRM1 | 3'UTR or downstream | 1.824 | - | - | 0.584 | GT | 1 | 1.08 | 0.0117 | 0.02938 | 0.61 | 0.20166 | 0.72 | 0.133223 | 0.76 | 0.111546 | Quillen et al., 2012 |
| 7 | 55109177 | rs12668421 | A | T | EGFR | intronic | 0.212 | - | - | 0.494 | IMP | 0.98 | -0.16 | 0.7473 | 0.7809 | -0.39 | 0.46139 | -0.14 | 0.787191 | -0.15 | 0.780554 | Quillen et al., 2012 |
| 7 | 55156071 | rs11238349 | G | A | EGFR | intronic | 0.431 | - | - | 0.393 | IMP | 1 | 0.34 | 0.48 | 0.542 | -0.65 | 0.22091 | -0.37 | 0.479127 | -0.34 | 0.523937 | Quillen et al., 2012 |
| 7 | 55454267 | rs4948023 | G | A | LANCL2 near EGFR | intronic | 4.667 | - | - | 0.684 | IMP | 1 | 0.16 | 0.7641 | 0.7956 | -0.22 | 0.696131 | 0.00 | 0.996604 | 0.00 | 0.994267 | Quillen et al., 2012 |
| 9 | 12682663 | rs10809826 | C | G | TYRP1 | upstream | 1.738 | - | - | 0.117 | IMP | 0.96 | -0.28 | 0.6803 | 0.7221 | -0.55 | 0.472775 | -0.65 | 0.392833 | -0.73 | 0.332409 | Adhikari et al., 2019 |
| 9 | 16864521 | rs2153271 | C | T | BNC2 | intronic | 20.5 | - | - | 0.152 | GT | 1 | -2.27 | 0.0002337 | 0.001459 | -2.29 | 0.00110657 | -2.25 | 0.00120438 | -2.37 | 0.000065015 | Ju and Mathieson, 2021 |
| 10 | 119564143 | rs111198112 | C | T | near EMX2 | intergenic | 17.79 | - | - | 0.187 | GT | 1 | 0.33 | 0.5664 | 0.6206 | 1.10 | 0.067385 | 0.99 | 0.100263 | 0.96 | 0.11598 | Adhikari et al., 2019 |
| 11 | 88511524 | rs7118677 | G | T | GRM5 near TYR | intronic | 2.034 | - | - | 0.144 | IMP | 1 | -2.42 | 0.0001392 | 0.000982 | -1.62 | 0.0152453 | -1.99 | 0.00286815 | -2.10 | 0.00151048 | Adhikari et al., 2019 |
| 11 | 88911696 | rs1042602 | C | A | TYR | S192Y | 23.8 | probably_damaging(0.974) | deleterious(0.01) | 0.072 | IMP | 0.74 | -2.90 | 0.0003701 | 0.002074 | -2.16 | 0.0119931 | -2.43 | 0.00469847 | -2.56 | 0.00273999 | Stokowski et al., 2007 |
| 11 | 89011046 | rs1393350 | G | A | TYR | intronic | 1.555 | - | - | 0.019 | GT | 1 | -3.51 | 0.01947 | 0.04354 | -2.90 | 0.0869741 | -3.38 | 0.042737 | -3.57 | 0.0261186 | Liu et al., 2015 |
| 11 | 89017961 | rs1126809 | G | A | TYR | R402Q | 27.2 | probably_damaging(0.994) | - | 0.019 | IMP | 0.97 | -3.51 | 0.01947 | 0.04354 | -2.90 | 0.0869741 | -3.38 | 0.042737 | -3.57 | 0.0261186 | Adhikari et al., 2019; Ju and Mathieson, 2021 |
| 12 | 89299746 | rs642742 | C | T | KITLG | upstream | 14.92 | - | - | 0.569 | GT | 1 | -0.34 | 0.4757 | 0.538 | -0.20 | 0.708049 | -0.30 | 0.568757 | -0.28 | 0.602548 | Sturm, 2009 |
| 12 | 89328335 | rs12821256 | T | C | KITLG | upstream | 15.74 | - | - | 0.015 | GT | 1 | 1.20 | 0.5327 | 0.5902 | -1.36 | 0.524421 | -1.08 | 0.608744 | -0.95 | 0.648883 | Ju and Mathieson, 2021 |
| 14 | 92773663 | rs12896399 | G | T | LOC105370627 near SLC24A4 | intronic | 0.043 | - | - | 0.054 | GT | 1 | -0.16 | 0.8692 | 0.887 | -0.96 | 0.352238 | -0.89 | 0.380027 | -0.98 | 0.337781 | Sulem et al., 2007 |
| 15 | 28197037 | rs1800414 | T | C | OCA2 | H615R | 23.3 | benign(0.133) | deleterious(0) | 0.070 | IMP | 0.26 | 0.48 | 0.6116 | 0.6612 | -0.01 | 0.989592 | 0.07 | 0.939843 | 0.08 | 0.938174 | Edwards et al., 2010 |
| 15 | 28213850 | rs4778219 | C | T | OCA2 | intronic | 1.527 | - | - | 0.316 | GT | 1 | -0.53 | 0.3074 | 0.3782 | -0.89 | 0.0952479 | -0.93 | 0.083945 | -0.90 | 0.0977717 | Adhikari et al., 2019 |
| 15 | 28235773 | rs1800404 | C | T | OCA2 | synonymous coding | 0.321 | - | - | 0.488 | GT | 1 | -1.50 | 0.001446 | 0.005889 | -1.47 | 0.00324612 | -1.30 | 0.00924527 | -1.37 | 0.00639755 | Crawford et al., 2017; Adhikari et al., 2019 |
| 15 | 28260053 | rs1800401 | G | A | OCA2 | R305W | 22.7 | benign(0.245) | deleterious(0.03) | 0.068 | GT | 1 | 0.03 | 0.9788 | 0.9814 |  |  |  |  |  |  | ... |
| 15 | 28344238 | rs7495174 | A | G | OCA2 | intronic | 7.622 | - | - | 0.087 | GT | 1 | 1.64 | 0.0326 | 0.0649 | 1.57 | 0.0663265 | 1.60 | 0.0580815 | 1.60 | 0.0567887 | Han et al., 2008 |

*Appendix 2—table 1 continued on next page*

Appendix 2—table 1 continued

| CHR | pos (b37) | SNP | REF | ALT | gene | location | CADD_PHRED | Polyphen (main) | SIFT (main) | Freq | GT source | AR2 | BETA_a | P_a_raw | P_a_adj | BETA_b | P_b | BETA_c | P_c | BETA_d | P_d | reference(s) |
|---|---|---|---|---|---|---|---|---|---|---|---|---|---|---|---|---|---|---|---|---|---|---|
| 15 | 28365618 | rs12913832 | A | G | HERC2 near OCA2 | intronic | 15.8 | - | - | 0.074 | GT | 1 | -1.86 | 0.03926 | 0.07497 | -1.48 | 0.112546 | -1.53 | 0.100514 | -1.57 | 0.0919142 | Liu et al., 2015; Adhikari et al., 2019 |
| 15 | 28380518 | rs4778249 | T | A | HERC2 near OCA2 | intronic | 0.649 | - | - | 0.790 | IMP | 1 | -2.23 | 0.0002214 | 0.0014 | -2.43 | 0.00041828 | -2.14 | 0.00162008 | -2.12 | 0.0016832 | Adhikari et al., 2019 |
| 15 | 28530182 | rs1667394 | C | T | HERC2 near OCA2 | intronic | 1.111 | - | - | 0.452 | GT | 1 | -1.42 | 0.002039 | 0.007666 | -1.56 | 0.00151414 | -1.41 | 0.0038335 | -1.43 | 0.00369895 | Sulem et al., 2007 |
| 16 | 89986117 | rs1805007 | C | T | MC1R | R151C | 25.2 | probably_damaging(0.996) | deleterious(0.02) | 0.016 | GT | 1 | 1.32 | 0.4775 | 0.5397 | 0.67 | 0.712981 | 0.93 | 0.613105 | 0.86 | 0.64501 | Ju and Mathieson, 2021 |
| 16 | 89986154 | rs885479 | G | A | MC1R | R163Q | 10.89 | benign(0.013) | tolerated(0.3) | 0.461 | IMP | 0.92 | -1.31 | 0.008565 | 0.0231 | -1.60 | 0.00241496 | -1.46 | 0.00572561 | -1.48 | 0.00546969 | Liu et al., 2015 |
| 19 | 3548231 | rs2240751 | A | G | MFSD12 | Y182H | 27.4 | probably_damaging(0.999) | deleterious(0) | 0.031 | GT | | -3.03 | 0.03735 | 0.0723 | -1.60 | 0.281792 | -1.46 | 0.33655 | -1.57 | 0.3027 | Adhikari et al., 2019 |
| 20 | 3625436 | rs562926 | C | T | ATRN | intronic or downstream | 4.601 | - | - | 0.402 | GT | 1 | 0.85 | 0.08705 | 0.1395 | -0.12 | 0.821567 | 0.18 | 0.730685 | 0.28 | 0.595369 | Quillen et al., 2012 |
| 20 | 32856998 | rs6058017 | A | G | ASIP/AHCY | 3'UTR/intron | 7.639 | - | - | 0.342 | IMP | 0.95 | -0.90 | 0.07274 | 0.1212 | -0.85 | 0.126901 | -1.04 | 0.0606301 | -1.04 | 0.0609305 | Stokowski et al., 2007 |

**Appendix 3**

## Supplementary Tables

**Appendix 3—table 1.** Top novel variants that may contribute towards skin pigmentation from our GWAS analysis. While the lowest p-values from the LMM-based methods meet the conventional criterion of 5e-08 for genome wide significance, the low observed minor allele frequencies (<2%) are inconsistent with what would be expected for variants responsible for pigmentation differences between the African and Native American populations.

| CHR | pos (b37) | SNP | REF | ALT | gene | location | CADD_PHRED | Freq | GT source | AR2 | BETA_a | P_a_raw | P_a_adj | BETA_b | P_b | BETA_c | P_c | BETA_d | P_d |
|---|---|---|---|---|---|---|---|---|---|---|---|---|---|---|---|---|---|---|---|
| 1 | 114560208 | rs113236485 | A | G | near SYT6 | intergenic | 1.323 | 0.014 | IMP | 0.91 | 11.96 | 1.01E-08 | 6.71E-07 | 12.24 | 1.57E-07 | 12.91 | 4.67E-08 | 12.96 | 2.17E-08 |
| 1 | 114576742 | rs145925324 | G | A | near SYT6 | intergenic | 0.648 | 0.013 | IMP | 1 | 12.50 | 1.09E-08 | 7.12E-07 | 12.56 | 1.55E-07 | 13.39 | 3.85E-08 | 13.49 | 1.81E-08 |
| 1 | 114581335 | rs141998140 | G | T | near SYT6 | intergenic | 2.099 | 0.013 | IMP | 1 | 12.50 | 1.09E-08 | 7.12E-07 | 12.56 | 1.55E-07 | 13.39 | 3.85E-08 | 13.49 | 1.81E-08 |
| 1 | 114582335 | rs187318390 | C | T | near SYT6 | intergenic | 1.165 | 0.013 | IMP | 1 | 12.50 | 1.09E-08 | 7.12E-07 | 12.56 | 1.55E-07 | 13.39 | 3.85E-08 | 13.49 | 1.81E-08 |
| 1 | 114586703 | rs149623066 | A | G | near SYT6 | intergenic | 0.052 | 0.013 | IMP | 1 | 12.50 | 1.09E-08 | 7.12E-07 | 12.56 | 1.55E-07 | 13.39 | 3.85E-08 | 13.49 | 1.81E-08 |
| 1 | 114595150 | rs78273840 | C | T | near SYT6 | intergenic | 1.805 | 0.017 | IMP | 0.78 | 8.87 | 3.13E-06 | 5.39E-05 | 10.59 | 5.74E-07 | 11.11 | 1.98E-07 | 11.25 | 9.38E-08 |
| 1 | 114611620 | rs116218201 | T | G | near SYT6 | intergenic | 2.199 | 0.012 | IMP | 0.83 | 14.09 | 1.12E-09 | 1.24E-07 | 14.18 | 3.11E-08 | 15.52 | 3.16E-09 | 15.69 | 1.23E-09 |
| 1 | 114612965 | rs116746819 | G | A | near SYT6 | intergenic | 0.351 | 0.012 | IMP | 0.84 | 14.09 | 1.12E-09 | 1.24E-07 | 14.18 | 3.11E-08 | 15.52 | 3.16E-09 | 15.69 | 1.23E-09 |
| 1 | 114614230 | rs549514340 | T | C | near SYT6 | intergenic | 3.308 | 0.012 | IMP | 0.84 | 14.09 | 1.12E-09 | 1.24E-07 | 14.18 | 3.11E-08 | 15.52 | 3.16E-09 | 15.69 | 1.23E-09 |
| 1 | 114619521 | rs115102845 | C | G | SYT6 | downstream | 0.253 | 0.012 | IMP | 0.86 | 14.09 | 1.12E-09 | 1.24E-07 | 14.18 | 3.11E-08 | 15.52 | 3.16E-09 | 15.69 | 1.23E-09 |
| 1 | 114620469 | rs182159269 | G | A | SYT6 | downstream | 0.225 | 0.012 | IMP | 0.85 | 14.09 | 1.12E-09 | 1.24E-07 | 14.18 | 3.11E-08 | 15.52 | 3.16E-09 | 15.69 | 1.23E-09 |
| 1 | 114620940 | rs183827287 | C | T | SYT6 | downstream | 0.12 | 0.012 | IMP | 0.92 | 14.09 | 1.12E-09 | 1.24E-07 | 14.18 | 3.11E-08 | 15.52 | 3.16E-09 | 15.69 | 1.23E-09 |
| 1 | 114622601 | rs141251595 | G | A | SYT6 | downstream | 2.003 | 0.012 | IMP | 0.93 | 14.09 | 1.12E-09 | 1.24E-07 | 14.18 | 3.11E-08 | 15.52 | 3.16E-09 | 15.69 | 1.23E-09 |
| 1 | 114624033 | rs186173861 | T | C | SYT6 | downstream | 2.87 | 0.012 | IMP | 0.99 | 14.09 | 1.12E-09 | 1.24E-07 | 14.18 | 3.11E-08 | 15.52 | 3.16E-09 | 15.69 | 1.23E-09 |
| 1 | 114624337 | rs181299762 | C | T | SYT6 | downstream | 1.726 | 0.012 | IMP | 0.99 | 14.09 | 1.12E-09 | 1.24E-07 | 14.18 | 3.11E-08 | 15.52 | 3.16E-09 | 15.69 | 1.23E-09 |
| 1 | 114625992 | rs111898196 | T | C | SYT6 | downstream | 14.83 | 0.012 | IMP | 0.98 | 14.09 | 1.12E-09 | 1.24E-07 | 14.18 | 3.11E-08 | 15.52 | 3.16E-09 | 15.69 | 1.23E-09 |
| 1 | 114626277 | rs546818700 | T | A | SYT6 | downstream | 0.549 | 0.012 | IMP | 0.98 | 14.09 | 1.12E-09 | 1.24E-07 | 14.18 | 3.11E-08 | 15.52 | 3.16E-09 | 15.69 | 1.23E-09 |
| 1 | 114626295 | rs112626676 | C | T | SYT6 | downstream | 6.822 | 0.012 | IMP | 0.98 | 14.09 | 1.12E-09 | 1.24E-07 | 14.18 | 3.11E-08 | 15.52 | 3.16E-09 | 15.69 | 1.23E-09 |
| 1 | 114656630 | rs185469828 | C | T | SYT6 | intronic | 3.735 | 0.011 | IMP | 1 | 13.32 | 7.49E-08 | 3.12E-06 | 13.55 | 1.61E-06 | 15.25 | 1.58E-07 | 15.44 | 6.93E-08 |
| 1 | 114656848 | rs79537167 | G | T | SYT6 | intronic | 0.359 | 0.011 | IMP | 1 | 13.32 | 7.49E-08 | 3.12E-06 | 13.55 | 1.61E-06 | 15.25 | 1.58E-07 | 15.44 | 6.93E-08 |
| 3 | 124343256 | rs676091 | T | G | KALRN | intronic | 3.169 | 0.017 | IMP | 0.99 | 9.59 | 3.60E-08 | 1.78E-06 | 9.73 | 8.49E-08 | 9.02 | 6.00E-07 | 9.21 | 2.08E-07 |
| 4 | 65768709 | rs6816819 | T | G | LOC107986284 | intronic | 5.827 | 0.013 | GT | 1 | 11.99 | 6.11E-10 | 7.84E-08 | 10.96 | 1.14E-07 | 10.30 | 5.30E-07 | 10.50 | 1.66E-07 |

key, a = linear regression, 10 PCs; b = LMM with 0 PCs, std GRM; c = LMM with 10 PCs, std GRM; d = LMM with 10 PCs, reap GRM; adj = based on lambda, inflation factor; beta = effect size.

