## [Editor Report]

This pigmentation study focuses on a community from Kalinago Territory from the Caribbean islands that on average possess high percentages of Indigenous American ancestry, and broadens the effort of quantifying the genetic effects on skin pigmentation in humans. This paper describes an analysis of the genetic structure of the Kalinago population in the Commonwealth of Dominica, and the relationship between ancestry and skin pigmentation in that population. They provide valuable new insights into the skin-lightening effect of Native American alleles, which likely have been obscured by the effect of European alleles in previous studies of admixed Native American populations. Additionally, this paper provides an interesting analysis of previously reported albinism alleles, which paints a more complex picture of the genetic architecture of pigmentation.

---

## [Decision Letter]

**Decision letter after peer review:**

Thank you for submitting your article "Native American Ancestry and Pigmentation Allele Contributions to Skin Color in a Caribbean Population" for consideration by *eLife*. Your article has been reviewed by 3 peer reviewers, and the evaluation has been overseen by a Reviewing Editor and Satyajit Rath as the Senior Editor. The reviewers have opted to remain anonymous.

The reviewers have discussed their reviews with one another, and the Reviewing Editor has drafted this to help you prepare a revised submission. We believe that the cohort and sampling effort is interesting and admirable and would help to diversify genomic studies and studies of skin pigmentation. However, the reviewers have found substantial issues with respect to the technical details of the analyses, the magnitude of the advance reported for the field, the availability of the data, and the consideration of previous studies and justification of analysis choices made. The authors will need to consider and respond to each of the reviewer points in a revised version for the manuscript to be considered further.

Essential revisions:

1) Why does the paper only consider the possibility for Native American-derived pigmentation alleles to contribute by decreasing pigmentation? The paper frames Native American and East Asian ancestry as necessarily tied to lighter pigmentation. However, this does not consider that there have been previous reports of Native American-European populations where Native American ancestry has been associated with darker skin. The authors need to offer a justification of the search for depigmentation alleles only or extend their analysis

2) The authors need to provide more clear details of all the analyses performed such as ADMIXTURE and association analysis and re-do these analyses as necessary given the technical points raised by reviewers #2 and #3. These include choice of K in the admixture analyses, potential new analyses of local ancestry to add rigor, considerations of relatedness in the association analysis and how predictors are chosen.

3) The paper should credit and include previous studies and hypotheses on skin pigmentation in different ancestries as mentioned by all reviewers, and update their models as necessary to consider more genes or justify their choices.

4) The authors should try to identify the specific NAM-specific variants that contribute to pigmentation either through GWAS or admixture mapping. If they can do that and actually discover new variants it would be a more novel contribution, and make the paper more likely to be eventually accepted for publication.

5) The authors need to provide a clearer data availability framework. If we understand correctly the data cannot be shared publicly due to community stipulations, but they can be shared at the discretion of the corresponding author with no further resort to the community? That seems inconsistent. Is this actually what is specified in the IRB and other agreements? Even if so, "reasonable request" is not well-defined and there needs to be a clear statement about under what conditions exactly the author commits to sharing the data. The authors need to share enough so that others can replicate their results, and there are ways they can do that without compromising privacy.

*Reviewer #1 (Recommendations for the authors):*

The authors should be consistent with the capitalization of 'Mixed' and 'Black.' Mixed is sometimes capitalized and sometimes not, and if there is no reason for the lowercase in 'black,' it should be capitalized to denote ethnicity as with the other terms. In the comparison of Kalinago vs. Mixed individuals, no reason is given to exclude those who identified as Black. Please provide an explanation or include them in the ancestry comparisons --this would only strengthen your point if African ancestry is higher in those groups.

Also, it would be helpful if there was some explanation as to how individuals were socially categorized (what phenotypes did people use to explain this? Was there any relation to parental identity?).

At one point, the authors describe the albino individuals as having "golden blonde" hair. There is no reason to add "golden"; it is not descriptive and gives the impression of superior valuation of this phenotype over others.

If possible, the authors should consider adding age and sex to the phenotype txt file.

Lastly, the authors need to justify why they only looked for albinism variants that were homozygous in the albino individual and heterozygous in the parent. I understand that this is a recessive trait; however, they discuss the allele they identify as having produced the phenotype in a compound heterozygote, so it bears elaborating on their reasoning for only considering homozygous variants in the Kalinago.

*Reviewer #2 (Recommendations for the authors):*

I thought the identification of the putative albinism allele was interesting and I don't have any issues with that analysis (would be nice to see an in vivo model eg in zebrafish but I certainly wouldn't require it for this paper).

I think the result that AFR ancestry is associated with higher MI relative to NAM is very unsurprising, unless I am missing something. It seems like the obvious next step would be to try to identify specific variants and I don't understand why the authors didn't do that.

I'm not totally convinced by the ancestry analysis, at least quantitatively. In general, performing local ancestry inference would enable additional analyses and more accurate approaches to some of the analyses you do perform (for example ancestry inference)

1) I don't think you can necessarily interpret quantitatively ADMIXTURE components in the way you want to. In general "ancestry" in that sense is not a very well-defined concept. For example, at K=4 what you describe as the "Native American component" is indeed maximized in Native American populations, but also present in East Asian populations. Similarly, the yellow ancestry cluster could equally well be described as "Middle Eastern" (yellow+black) rather than "European" (yellow+green) ancestry based on your admixture plots. In fact, some of your African populations actually have blacK^+^yellow ancestry. So you could equally interpret this as the Kalinago having NAM ancestry, and ancestry from some unsampled (e.g. East) African population that has about 70/30 of the black/yellow component. That would also explain the high correlation between your inferred African and European ancestry in Figure 2A.

I think the reason you call the yellow component "European", is because that's what you expect to see based on your knowledge of recent history. So you mean ancestry in sense of recent admixture, which is probably better-defined. But in that case, I do not think you can rely on ADMIXTURE to give you unbiased estimates. Local ancestry-based methods would probably be better and would give you confidence based on the tract length distribution that you are actually detecting recent European admixture.

2) Even if there is recent European ancestry, the two variants at SLC45A2 and SLC45A2 are the largest-effect European pigmentation alleles but they are by no means the only ones. For example, there are over 100 genomes-wide significant associations with skin pigmentation in UK Biobank, many of which have been reported as targets of selection and/or as contributing to differences in pigmentation between European and other populations. Some examples include TYRP1 (PMID: 17233754), TYR (24616518), KITLG (27738015), MC1R (24045876), as well as other variants at OCA2/HERC (17182896). This is not an exhaustive list of genes or references for each gene but you get the idea. Therefore, I don't think it's necessarily the case that including the two large-effect alleles will account for all the effect of European ancestry in your analysis. That said, since EUR ancestry is negatively correlated with NAM ancestry in your sample, I don't think that qualitatively affects the result.

3) But even ignoring the European ancestry, the framing is a bit odd. Figure 5 shows that NAM ancestry is associated with lower MI or, equivalently that higher AFR ancestry is associated with higher MI. But isn't this already known. We know that (even relatively unadmixed) Native American populations have lighter skin pigmentation than West African populations, so don't we expect that? Or are you claiming that we do not know that already?

4) It seems that the next step would be to try to identify the specific NAM-specific variants that contribution to pigmentation either through GWAS or admixture mapping. If you could do that and actually discover new variants it would be a more novel contribution.

5) You have a very large age range in this study and should probably include age and age^2 at least as covariates in the regression models to account for age-related variation in pigmentation (which may be confounded with age-related changes in ancestry). I'd actually question whether children should be included at all, both from a scientific (and perhaps also an ethical) perspective.

6) I am a bit confused about the data availability. If I understand correctly the data cannot be shared publicly due to community stipulations, but they CAN be shared at the discretion of the corresponding author with no further resort to the community? That seems inconsistent. Is this actually what is specified in the IRB and other agreements? Even if so, "reasonable request" is not well-defined and there needs to be a clear statement about under what conditions exactly the author commits to sharing the data. If the data actually cannot be shared, then say so.

*Reviewer #3 (Recommendations for the authors):*

1. The introduction has claimed that "the genetic basis for lighter skin pigmentation in Native American and East Asian populations…has yet to be established". This is misleading as there have been studies on the convergent evolution and shared genetic basis of light skin between East Asians and Europeans on KITLG, and East Asian specific allele in OCA2 (Beleza et al., 2012; Yang et al., 2016); variants on lighter skin in Native Americans in MFSD12 (Adhikari et al., 2019), and selection-test suggested OPRM1 and EGFR (Quillen et al., 2011), to name a few. Perhaps a starting place to gather the relevant literature would be Quillen et al., 2018.

This brings in two additional questions for this study that the authors might want to clarify in the manuscript:

(1) The significance of this study in the presence of other pigmentation studies in individuals with Native American ancestry, especially with so few genes that this study highlighted that were already assessed in other Latin American cohorts. This is perhaps something the authors should put into the discussion.

(2) The rationale behind choosing only three variants from three genes (SLC24A5, SLC45A2, OCA2) as covariates to characterize their effects in.

2. The ancestry decomposition was done using ADMIXTURE, while a lot of details were lacking for sufficient judgment and independent replication. The necessary information includes: What was the final K used to obtain the "Native American" ancestry estimate used in all subsequent analyses (presumably K=6?), and how was it determined (e.g. cross validation)? How many iterations with random seeds were used to obtain the estimates, and how many minor modes appeared? Was data properly pruned based on linkage disequilibrium, and if so what was the threshold and how many variants were left for ADMIXTURE? What is the maximal K ever tested? If the authors used k=6 to obtain the indigenous ancestry estimate, K>6 should be run to make sure this is the stable performance and reasonable decision.

3. The authors acknowledged presence of close relatedness, which can introduce bias in lot of analyses, unless properly handled. The description is lacking here, for example, in association analyses, were close relatedness pruned or a GRM is used in the model? If the authors merely partitioned relatedness into different groups for separate GWAS, then meta-analyzed the summary stats from all groups, this is still equivalent to including every sample into one GWAS. Moreover, the kinship threshold the authors used to partition samples seemed relaxed (i.e. to exclude second degree relatedness), when cryptic relatedness can largely present, since it was a community recruitment for the participants in this study with 15% of the total population involved. I would encourage the authors to try using a standard linear mixed model to account for relatedness, if not done so, and examining the genome inflation factor for indication of any leftover structures.

4. The association analysis: There are no summary stats reported, or any conclusions found in the manuscript with regard to the result of GWAS: even if no variants are significant given the limitation in power, this needs to be reported. Additionally, the authors might consider reporting the effect size and p values for variants in the suggestive Native American specific pigmentation genes from previous literatures (see point 1).

5. The predictors in the linear model: how are the predictors chosen is unclear, why does a discovery GWAS needs to account for specifically three variants from SLC24A5, SLC45A2, and OCA2, not none, less, or more? The participants included children, as young as below 10, thus was age tested for a significant covariate in this case? (similar to point 2) what ancestries are included as covariates, and which K were the estimates derived from?

6. The effect size of OCA2NW273KV as -8 MI: this is from the discovery cohort, and no replication of the effect size is done. This is worth bringing into discussion for winner's curse effect.

7. Line 107 above Figure 2: relatively recent divergence between any two populations does not necessarily suggest sharing of any causal alleles of a particular trait. This sentence reads as an overstatement.

8. Line 217 after Figure 3: The authors mentioned the non-albino individuals with lighter pigmentation, yet their quantitative description of the phenotype is lacking. What is their hair color, and MI of skin that make them classified as "lighter" than the rest of the samples, leading to the further analysis of their genotypes?

9. Line 243 above Table 2: The excess of 3% allele frequency in SLC24A5 compared to global European ancestry is a very small amount for the authors to claim that there is another source of ancestry to bring in the derived allele. 3% can easily be a stochastic /estimation error from global ancestry.

[Editors’ note: further revisions were suggested prior to acceptance, as described below.]

Thank you for resubmitting your work entitled "Native American Ancestry and Pigmentation Allele Contributions to Skin Color in a Caribbean Population" for further consideration by *eLife*. Your revised article has been evaluated by Satyajit Rath (Senior Editor) and a Reviewing Editor.

The manuscript has been improved but there are some remaining issues that need to be addressed, as outlined below:

– The authors are urged to do more critical/thorough engagement with how people are socially categorized in the study. The authors should review recent guidelines highlighted below and take the opportunity to make their manuscript (title, abstract, main text) in line with guidelines.

NASEM report:

https://nap.nationalacademies.org/catalog/26902/using-population-descriptors-in-genetics-and-genomics-research-a-new

Safra Center working group recommendations:

https://www.science.org/doi/10.1126/science.abm7530

Attached ethical framework for research that uses genetic ancestry (soon to be in print)

– The authors should check their citations as they have one author (Jada Benn Torres) down as both Torres JB and Benn-Torres, J (it should be Benn Torres, J.)

– The authors should make the full GWAS summary statistics public, either in the supplement or on the GWAS catalog. Since the raw data are not available, it's important to release the summary stats for others to build on e.g. with meta analysis or other analysis.

---

## [Author Response]

Essential revisions:Reviewer #1 (Recommendations for the authors):The authors should be consistent with the capitalization of 'Mixed' and 'Black.' Mixed is sometimes capitalized and sometimes not, and if there is no reason for the lowercase in 'black,' it should be capitalized to denote ethnicity as with the other terms. In the comparison of Kalinago vs. Mixed individuals, no reason is given to exclude those who identified as Black. Please provide an explanation or include them in the ancestry comparisons --this would only strengthen your point if African ancestry is higher in those groups.

Thank you for the comment. We have now consistently capitalized Black and Mixed. Black individuals were not excluded from analysis; however, none of the small number of individuals meeting this description were sampled.

Also, it would be helpful if there was some explanation as to how individuals were socially categorized (what phenotypes did people use to explain this? Was there any relation to parental identity?).

They were community-defined based on their skin color, but this subject was not explored further in our interviews.

At one point, the authors describe the albino individuals as having "golden blonde" hair. There is no reason to add "golden"; it is not descriptive and gives the impression of superior valuation of this phenotype over others.

Thank you. We removed the word ‘golden’.

If possible, the authors should consider adding age and sex to the phenotype txt file.

We included the phenotype and sex to the text file (PC_Covariates_Age_Sex.xlxs) found in Dryad.

Lastly, the authors need to justify why they only looked for albinism variants that were homozygous in the albino individual and heterozygous in the parent. I understand that this is a recessive trait; however, they discuss the allele they identify as having produced the phenotype in a compound heterozygote, so it bears elaborating on their reasoning for only considering homozygous variants in the Kalinago.

Thank you for pointing out this. We changed our wording to make this clearer. Had we not found a homozygous variant, we would have proceeded to examine the sequence data for heterozygous variants.

Reviewer #2 (Recommendations for the authors):I thought the identification of the putative albinism allele was interesting and I don't have any issues with that analysis (would be nice to see an in vivo model eg in zebrafish but I certainly wouldn't require it for this paper).

Thank you. We agree that in vivo model will be nice and will be pursuing it separately from the focus of this manuscript.

I think the result that AFR ancestry is associated with higher MI relative to NAM is very unsurprising, unless I am missing something. It seems like the obvious next step would be to try to identify specific variants and I don't understand why the authors didn't do that.

GWAS results are reported in more detail than before. No novel variant was significant in our sample.

I'm not totally convinced by the ancestry analysis, at least quantitatively. In general, performing local ancestry inference would enable additional analyses and more accurate approaches to some of the analyses you do perform (for example ancestry inference)1) I don't think you can necessarily interpret quantitatively ADMIXTURE components in the way you want to. In general "ancestry" in that sense is not a very well-defined concept. For example, at K=4 what you describe as the "Native American component" is indeed maximized in Native American populations, but also present in East Asian populations. Similarly, the yellow ancestry cluster could equally well be described as "Middle Eastern" (yellow+black) rather than "European" (yellow+green) ancestry based on your admixture plots. In fact, some of your African populations actually have blacK^+^yellow ancestry. So you could equally interpret this as the Kalinago having NAM ancestry, and ancestry from some unsampled (e.g. East) African population that has about 70/30 of the black/yellow component. That would also explain the high correlation between your inferred African and European ancestry in Figure 2A.I think the reason you call the yellow component "European", is because that's what you expect to see based on your knowledge of recent history. So you mean ancestry in sense of recent admixture, which is probably better-defined. But in that case, I do not think you can rely on ADMIXTURE to give you unbiased estimates. Local ancestry-based methods would probably be better and would give you confidence based on the tract length distribution that you are actually detecting recent European admixture.

We are aware that it can be tempting to over-interpret the outputs from Admixture. For example, the appearance of the 'Native American' component in some East Asian populations does not man that these groups have actual Native American ancestry. Thank you for pointing out that we have described the clusters in ways that could be misleading. We changed our description to referring to the clusters by color. We have also added comparison to the results of local ancestry analysis, which are largely concordant.

2) Even if there is recent European ancestry, the two variants at SLC45A2 and SLC45A2 are the largest-effect European pigmentation alleles but they are by no means the only ones. For example, there are over 100 genomes-wide significant associations with skin pigmentation in UK Biobank, many of which have been reported as targets of selection and/or as contributing to differences in pigmentation between European and other populations. Some examples include TYRP1 (PMID: 17233754), TYR (24616518), KITLG (27738015), MC1R (24045876), as well as other variants at OCA2/HERC (17182896). This is not an exhaustive list of genes or references for each gene but you get the idea. Therefore, I don't think it's necessarily the case that including the two large-effect alleles will account for all the effect of European ancestry in your analysis. That said, since EUR ancestry is negatively correlated with NAM ancestry in your sample, I don't think that qualitatively affects the result.

We report effects (if any) of previously reported pigmentation-associated variants (Appendix 2 – Table 6).

3) But even ignoring the European ancestry, the framing is a bit odd. Figure 5 shows that NAM ancestry is associated with lower MI or, equivalently that higher AFR ancestry is associated with higher MI. But isn't this already known. We know that (even relatively unadmixed) Native American populations have lighter skin pigmentation than West African populations, so don't we expect that? Or are you claiming that we do not know that already?

We have rearranged our results in a way that emphasizes that we are interested in the size of the pigmentation difference that remains to be explained.

4) It seems that the next step would be to try to identify the specific NAM-specific variants that contribution to pigmentation either through GWAS or admixture mapping. If you could do that and actually discover new variants it would be a more novel contribution.

We have now added Linear Mixed Models (LMM) based genome wide assessment to our earlier linear regression methods to identify NAM-specific variants. While the lowest p-values from the LMM-based methods meet the conventional criterion of 5e-08 for genome wide significance, the low observed minor allele frequencies (<2%) are inconsistent with what would be expected for variants responsible for pigmentation differences between the African and Native American populations. For example, a hypothetical allele that is fixed in Native Americans like *SLC24A5^A111T^* is in CEU would be expected to have a minor allele frequency close to the average percent Native American Ancestry in the Kalinago, about 0.6.

5) You have a very large age range in this study and should probably include age and age^2 at least as covariates in the regression models to account for age-related variation in pigmentation (which may be confounded with age-related changes in ancestry). I'd actually question whether children should be included at all, both from a scientific (and perhaps also an ethical) perspective.

We show in the revised manuscript that the effect of age on pigmentation is not significant, and that inclusion in the linear regression model has very little effect on parameter estimates (Appendix 1 – Table 5).

6) I am a bit confused about the data availability. If I understand correctly the data cannot be shared publicly due to community stipulations, but they CAN be shared at the discretion of the corresponding author with no further resort to the community? That seems inconsistent. Is this actually what is specified in the IRB and other agreements? Even if so, "reasonable request" is not well-defined and there needs to be a clear statement about under what conditions exactly the author commits to sharing the data. If the data actually cannot be shared, then say so.

We have updated our data availability statement.

Reviewer #3 (Recommendations for the authors):1. The introduction has claimed that "the genetic basis for lighter skin pigmentation in Native American and East Asian populations…has yet to be established". This is misleading as there have been studies on the convergent evolution and shared genetic basis of light skin between East Asians and Europeans on KITLG, and East Asian specific allele in OCA2 (Beleza et al., 2012; Yang et al., 2016); variants on lighter skin in Native Americans in MFSD12 (Adhikari et al., 2019), and selection-test suggested OPRM1 and EGFR (Quillen et al., 2011), to name a few. Perhaps a starting place to gather the relevant literature would be Quillen et al., 2018.This brings in two additional questions for this study that the authors might want to clarify in the manuscript:(1) The significance of this study in the presence of other pigmentation studies in individuals with Native American ancestry, especially with so few genes that this study highlighted that were already assessed in other Latin American cohorts. This is perhaps something the authors should put into the discussion.(2) The rationale behind choosing only three variants from three genes (SLC24A5, SLC45A2, OCA2) as covariates to characterize their effects in.

We have clarified that there are indeed genetic variants that contribute to skin color variation among East Asians and between East Asians or Native Americans and Africans, but none of the identified variants has an effect equivalent to that of the well-characterized variants found in Europeans. We have added (Appendix 2 – Table 6) reported p-values and effect sizes in our sample for previously reported variants.

2. The ancestry decomposition was done using ADMIXTURE, while a lot of details were lacking for sufficient judgment and independent replication. The necessary information includes: What was the final K used to obtain the "Native American" ancestry estimate used in all subsequent analyses (presumably K=6?), and how was it determined (e.g. cross validation)? How many iterations with random seeds were used to obtain the estimates, and how many minor modes appeared? Was data properly pruned based on linkage disequilibrium, and if so what was the threshold and how many variants were left for ADMIXTURE? What is the maximal K ever tested? If the authors used k=6 to obtain the indigenous ancestry estimate, K>6 should be run to make sure this is the stable performance and reasonable decision.

As we show, the ancestry estimates from K=4 to K=6 are very similar, since only a small number of individuals have substantial amounts of East Asian ancestry, and negligible amounts of Central/South Asian or Oceanian ancestry. Analysis at higher values of K does not improve our knowledge of Kalinago ancestry.

3. The authors acknowledged presence of close relatedness, which can introduce bias in lot of analyses, unless properly handled. The description is lacking here, for example, in association analyses, were close relatedness pruned or a GRM is used in the model? If the authors merely partitioned relatedness into different groups for separate GWAS, then meta-analyzed the summary stats from all groups, this is still equivalent to including every sample into one GWAS. Moreover, the kinship threshold the authors used to partition samples seemed relaxed (i.e. to exclude second degree relatedness), when cryptic relatedness can largely present, since it was a community recruitment for the participants in this study with 15% of the total population involved. I would encourage the authors to try using a standard linear mixed model to account for relatedness, if not done so, and examining the genome inflation factor for indication of any leftover structures.

We have removed the meta-analysis and added analysis using LMM-based models to the linear regression model. The decrease in λ using a simple GRM (and omitting PC-based covariates) suggests that this approach better accounts for relatedness. Models that include a GRM and add back the PC-based covariates exhibit greater statistic inflation that does not precisely follow the expected distribution.

4. The association analysis: There are no summary stats reported, or any conclusions found in the manuscript with regard to the result of GWAS: even if no variants are significant given the limitation in power, this needs to be reported. Additionally, the authors might consider reporting the effect size and p values for variants in the suggestive Native American specific pigmentation genes from previous literatures (see point 1).

We have added Appendix 2 – Table 6 to report the variants with the lowest p-values, and have displayed QQ-plots in Figure 5-Supplement Figure 2.

5. The predictors in the linear model: how are the predictors chosen is unclear, why does a discovery GWAS needs to account for specifically three variants from SLC24A5, SLC45A2, and OCA2, not none, less, or more? The participants included children, as young as below 10, thus was age tested for a significant covariate in this case? (similar to point 2) what ancestries are included as covariates, and which K were the estimates derived from?

Age has an effect that is not significant in our sample. We have included more detail on the effects of adding additional covariates such as age or other pigmentation-associated variants (Appendix 1 – Table 5).

6. The effect size of OCA2NW273KV as -8 MI: this is from the discovery cohort, and no replication of the effect size is done. This is worth bringing into discussion for winner's curse effect.

We agree that replication, preferably in a different population, would be useful. Such experiments lie outside the scope of the current work.

7. Line 107 above Figure 2: relatively recent divergence between any two populations does not necessarily suggest sharing of any causal alleles of a particular trait. This sentence reads as an overstatement.

We have removed this statement.

8. Line 217 after Figure 3: The authors mentioned the non-albino individuals with lighter pigmentation, yet their quantitative description of the phenotype is lacking. What is their hair color, and MI of skin that make them classified as "lighter" than the rest of the samples, leading to the further analysis of their genotypes?

We have clarified these points, including that the individuals described as lighter have m-index lower than 30.

9. Line 243 above Table 2: The excess of 3% allele frequency in SLC24A5 compared to global European ancestry is a very small amount for the authors to claim that there is another source of ancestry to bring in the derived allele. 3% can easily be a stochastic /estimation error from global ancestry.

We have revised this discussion considerably.

[Editors' note: further revisions were suggested prior to acceptance, as described below.]

The manuscript has been improved but there are some remaining issues that need to be addressed, as outlined below:– The authors are urged to do more critical/thorough engagement with how people are socially categorized in the study. The authors should review recent guidelines highlighted below and take the opportunity to make their manuscript (title, abstract, main text) in line with guidelines.NASEM report:https://nap.nationalacademies.org/catalog/26902/using-population-descriptors-in-genetics-and-genomics-research-a-newSafra Center working group recommendations:https://www.science.org/doi/10.1126/science.abm7530Attached ethical framework for research that uses genetic ancestry (soon to be in print)– The authors should check their citations as they have one author (Jada Benn Torres) down as both Torres JB and Benn-Torres, J (it should be Benn Torres, J.)– The authors should make the full GWAS summary statistics public, either in the supplement or on the GWAS catalog. Since the raw data are not available, it's important to release the summary stats for others to build on e.g. with meta analysis or other analysis.

As suggested by the editors, we have reworded appropriate parts of the paper to comply to the latest NASEM report. For example, we added the term ‘genetic’ or ‘genealogical’ before ancestry, as specific population descriptors.

As requested, we have also added GWAS summary statistic to GWAS-Catalog with the corresponding submission ID: https://www.ebi.ac.uk/gwas/deposition/submission/6467ad190627bb0001de9b55

We thank the editors for pointing out the error in the citations, which we have corrected accordingly.